# T-Rep: Representation Learning for Time Series using Time-Embeddings

**Archibald Fraikin**
Let it Care
PariSanté Campus, Paris, France
archibald.fraikin@inria.fr

**Adrien Bennetot**
Let it Care
PariSanté Campus, Paris, France
adrien.bennetot@letitcare.com

**Stéphanie Allassonnière**
Université Paris Cité, INRIA, Inserm, SU
Centre de Recherche des Cordeliers, Paris
stephanie.allassonniere@inria.fr

## Abstract

Multivariate time series present challenges to standard machine learning techniques, as they are often unlabeled, high dimensional, noisy, and contain missing data. To address this, we propose T-Rep, a self-supervised method to learn time series representations at a timestep granularity. T-Rep learns vector embeddings of time alongside its feature extractor, to extract temporal features such as trend, periodicity, or distribution shifts from the signal. These time-embeddings are leveraged in pretext tasks, to incorporate smooth and fine-grained temporal dependencies in the representations, as well as reinforce robustness to missing data. We evaluate T-Rep on downstream classification, forecasting, and anomaly detection tasks. It is compared to existing self-supervised algorithms for time series, which it outperforms in all three tasks. We test T-Rep in missing data regimes, where it proves more resilient than its counterparts. Finally, we provide latent space visualisation experiments, highlighting the interpretability of the learned representations.

## 1 Introduction

Multivariate time series have become ubiquitous in domains such as medicine, climate science, or finance. Unfortunately, they are high-dimensional and complex objects with little data being labeled (Yang & Wu, 2006), as it is an expensive and time-consuming process. Leveraging unlabeled data to build unsupervised representations of multivariate time series has thus become a challenge of great interest, as these embeddings can significantly improve performance in tasks like forecasting, classification, or anomaly detection (Deldari et al., 2021; Su et al., 2019). This has motivated the development of self-supervised learning (SSL) models for time series, first focusing on constructing instance-level representations for classification and clustering (Tonekaboni et al., 2021; Franceschi et al., 2019; Wu et al., 2018). More fine-grained representations were then developed to model time series at the timestep-level (Yue et al., 2022), which is key in domains such as healthcare or sensor systems. With fine-grained embeddings, one can capture subtle changes, periodic patterns, and irregularities that are essential for anomaly detection (Keogh et al., 2006) as well as understanding and forecasting disease progression. These representations can also be more resilient than raw data in the face of inter-sample variability or missing data (Yue et al., 2022), common issues in Human Activity Recognition (HAR) and medicine.

A central issue when learning representations of time series is the incorporation of time in the latent space, especially for timestep-level embeddings. In SSL, the temporal structure is learned thanks to the pretext tasks. In current state-of-the-art (SOTA) models, these tasks are contrastive (Tonekaboni et al., 2021; Yue et al., 2022; Banville et al., 2021), which poses important limitations (Zhang et al., 2023). In contrastive techniques, the learning signal is binary: positive pairs should be similar, while negative pairs should be very different (Chen et al., 2020). This makes obtaining a continuous or

fine-grained notion of time in the embeddings unfeasible, as these tasks only express *whether* two points should be similar, but not *how* close or similar they should be. Embedded trajectories are thus unlikely to accurately reflect the data's temporal structure.

Further, temporal contrastive tasks are incompatible with finite-state systems, where the signal transitions between $S$ states through time, regularly (periodic signal) or irregularly. Such tasks define positive pairs by proximity in time, and negative pairs by points that are distant in time (Banville et al., 2021; Franceschi et al., 2019), which can incur *sampling bias* issues. Points of a negative pair might be far in time but close to a period apart (i.e. very similar) and points of a positive pair might be close but very different (think of a pulsatile signal for example). This incoherent information hinders learning and may result in a poor embedding structure. Finite-state systems are extremely common in real-world scenarios such as sensor systems, medical monitoring, or weather systems, making the treatment of these cycles crucial.

To address the above issues, we propose T-Rep, a self-supervised method for learning fine-grained representations of (univariate and multivariate) time series. T-Rep improves the treatment of time in SSL thanks to the use of time-embeddings, which are integrated in the feature-extracting encoder and leveraged in the pretext tasks, helping the model learn detailed time-related features. We define as time-embedding a vector embedding of time, obtained as the output of a learned function $h_\psi$, which encodes temporal signal features such as trend, periodicity, distribution shifts etc. Time-embeddings thus enhance our model's resilience to missing data, and improve its performance when faced with finite-state systems and non-stationarity. We evaluate T-Rep on a wide variety of datasets in classification, forecasting and anomaly detection (see section 5), notably on Sepsis (Reyna et al., 2020a; Goldberger et al., 2000) a real-world dataset containing multivariate time series from $40,336$ patients in intensive care units (ICU), featuring noisy and missing data. Our major contributions are summarised as follows:

- To the best of our knowledge, we propose the first self-supervised framework for time series to leverage time-embeddings in its pretext tasks. This helps the model learn fine-grained temporal dependencies, giving the latent space a more coherent temporal structure than existing methods. The use of time-embeddings also encourages resilience to missing data, and produces more information-dense and interpretable embeddings.

- We compare T-Rep to SOTA self-supervised models for time series in classification, forecasting and anomaly detection. It consistently outperforms all baselines whilst using a lower-dimensional latent space, and also shows stronger resilience to missing data than existing methods. Further, our latent space visualisation experiments show that the learned embeddings are highly interpretable.

## 2 RELATED WORK

**Representation Learning for time series** The first techniques used in the field were encoder-decoder based, trained to reconstruct the original time series. Such models include Autowarp (Abid & Zou, 2018), TimeNet (Malhotra et al., 2017) and LSTM-SAE (Sagheer & Kotb, 2019), which all feature an RNN-based architecture. Variational Auto-Encoders (Kingma & Welling, 2013) inspired models have also been used, notably Interfusion (Li et al., 2021), SOM-VAE (Fortuin et al., 2018), and OmniAnomaly (Su et al., 2019) which combines a VAE with normalising flows. Encoder-only methods have been more popular recently, often based on contrastive approaches (Zhang et al., 2023). The Contrastive Predictive Coding (CPC) (Oord et al., 2018) framework tries to maximise the mutual information between future latent states and a context vector, using the infoNCE loss. This approach has been adapted for anomaly detection (Deldari et al., 2021) and general representations (Eldele et al., 2021). TS-TCC (Eldele et al., 2021) augments CPC by applying weak and strong transformations to the raw signal. Also, augmentation-based contrastive methods in computer vision (Chen et al., 2020) have been adapted to time series by changing the augmentations (Kiyasseh et al., 2021). Domain-specific transformations were proposed for wearable sensors (Cheng et al., 2020) and ECGs (Kiyasseh et al., 2021), such as noise injection, cropping, warping, and jittering (Pöppelbaum et al., 2022). The issue with these methods is they make transformation-invariance assumptions which may not be satisfied by the signal (Zhang et al., 2023; Yue et al., 2022). TS2Vec (Yue et al., 2022) addresses this with contextual consistency.

**Time-Embeddings in time series representations** have only been used in transformer-based architectures, which require a *positional encoding* module (Vaswani et al., 2017). While some use the original fixed sinusoidal positional encoding (Haresamudram et al., 2020; Zhang et al., 2022), Zerveas et al. (2021) and Tipirneni & Reddy (2022) chose to learn a time-embedding using a linear layer and a fully-connected layer respectively. Using or learning more sophisticated time-embeddings is a largely unexplored avenue that seems promising for dealing with long-term trends and seasonality (periodicity) in sequential data (Zhang et al., 2023; Wen et al., 2022). The most elaborate time-embedding for time series is Time2Vec (Kazemi et al., 2019), which was developed for supervised learning tasks and has not yet been exploited in a self-supervised setting. In existing self-supervised models, the time-embedding is used by the encoder to provide positional information (Zerveas et al., 2021; Tipirneni & Reddy, 2022; Haresamudram et al., 2020; Zhang et al., 2022), but is never exploited in pretext tasks. The best performing models use contrastive techniques to learn temporal dependencies, which only provide a binary signal (Zhang et al., 2023). T-Loss (Franceschi et al., 2019) follows the assumption that neighboring windows should be similar, and builds a triplet loss around this idea. TNC (Tonekaboni et al., 2021) extends this framework, dividing the signal into stationary windows to construct its positive and negative pairs. In Banville et al. (2021); Yue et al. (2022); Franceschi et al. (2019), positive and negative pairs are delimited by a threshold on the number of differing timesteps, which makes these methods unsuited to capturing periodic or irregularly recurring patterns in the data. Further, all these contrastive methods are quite coarse, making it hard to learn fine-grained temporal dependencies.

To summarise, existing methods have made tremendous progress in extracting spatial features from time series, but temporal feature learning is still limited. In particular, they are not suited to handling recurring patterns (periodic or irregular), and struggle to learn fine-grained temporal dependencies, because of the binary signal and sampling bias of contrastive tasks.

## 3 BACKGROUND

The aim of this work is to improve the treatment of time in representation learning for temporal data. These innovations are combined with state-of-the-art methods for spatial feature-learning and model training, the *contextual consistency* and *hierarchical loss* frameworks (Yue et al., 2022).

### 3.1 PROBLEM DEFINITION

Given a dataset $X = \{\mathbf{x}_1, ..., \mathbf{x}_N\} \in \mathbb{R}^{N \times T \times C}$ of $N$ time series of length $T$ with $C$ channels, the objective of self-supervised learning is to learn a function $f_\theta$, s.t. $\forall i \in [0, N]$, $\mathbf{z}_i = f_\theta(\mathbf{x}_i)$. Each $\mathbf{z}_i \in \mathbb{R}^{T \times F}$ is a representation of $\mathbf{x}_i$ of length $T$ and with $F$ channels, which should preserve as many features of the original data as possible. $f_\theta(\cdot)$ is learned by designing artificial supervised signals, called *pretext tasks*, from the unlabeled data $X$.

### 3.2 CONTEXTUAL CONSISTENCY

The objective of *contextual consistency* is to learn context-invariant representations of time series. The idea is to sample two overlapping segments $\mathbf{x}_1$ and $\mathbf{x}_2$ of the time series $\mathbf{x}$, to which random timestamp masking is applied, thus creating two different *contexts* (random choice of window and masked timesteps) for the overlapping window. Representations in the overlapping timesteps are then encouraged to be similar, leading to context-invariant representations. Two contrastive tasks were introduced by Yue et al. (2022) alongside the contextual consistency framework to extract spatial and temporal features.

**Instance-wise contrasting** encourages representations of the same time series under different contexts to be encoded similarly, and for different instances to be dissimilar (Yue et al., 2022). Let $B$ be the batch-size, $i$ the time series index and $t$ a timestep. $\mathbf{z}_{i,t}$ and $\mathbf{z}'_{i,t}$ denote the corresponding representation vectors under 2 different contexts. The loss function is given by:

$$\mathcal{L}_{inst}^{(i,t)} = -\log \frac{\exp\left(\mathbf{z}_{i,t} \cdot \mathbf{z}'_{i,t}\right)}{\sum_{j=1}^{B}\left(\exp\left(\mathbf{z}_{i,t} \cdot \mathbf{z}'_{j,t}\right) + \mathbb{1}_{i \neq j}\exp\left(\mathbf{z}_{i,t} \cdot \mathbf{z}_{j,t}\right)\right)}. \tag{1}$$

**Temporal contrasting** encourages representations of time series under different contexts to be encoded similarly when their respective timesteps match, and far apart when the timesteps differ (Yue et al., 2022). The loss function is given in Eq. 2, where $\Omega$ is the set of timesteps in the overlap between the 2 subseries:

$$\mathcal{L}_{temp}^{(i,t)} = -\log \frac{\exp\left(\mathbf{z}_{i,t} \cdot \mathbf{z}'_{i,t}\right)}{\sum_{t' \in \Omega}\left(\exp\left(\mathbf{z}_{i,t} \cdot \mathbf{z}'_{i,t'}\right) + \mathbb{1}_{t \neq t'} \exp\left(\mathbf{z}_{i,t} \cdot \mathbf{z}_{i,t'}\right)\right)}. \tag{2}$$

### 3.3 Hierarchical Loss

The hierarchical loss framework applies and sums the model's loss function at different scales, starting from a per-timestep representation and applying `maxpool` operations to reduce the time-dimension between scales (Yue et al., 2022). This gives users control over the granularity of the representation used for downstream tasks, without sacrificing performance. It also makes the model more robust to missing data, as it makes use of long-range information in the surrounding representations to reconstruct missing timesteps (Yue et al., 2022).

## 4 Method

### 4.1 Encoder Architecture

We present below our convolutional encoder, which contains 3 modules. The overall model structure is illustrated in Figure 1.

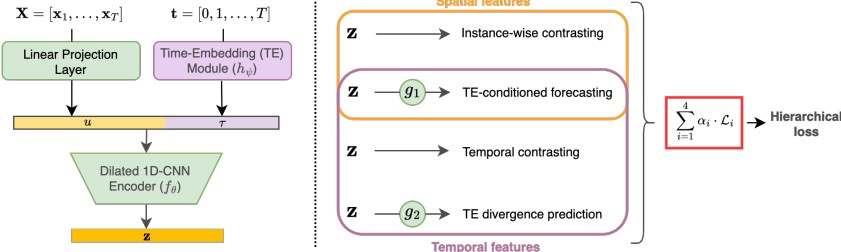

Figure 1: T-Rep architecture and workflow. The left part shows how the different modules interact (linear projection, time-embedding module, and encoder). The middle part shows the 4 pretext tasks used to train the model, and the kind of features they extract. The right hand side shows the loss computation: a linear combination of individual pretext task losses is passed to the *hierarchical loss* algorithm (see Appendix A.2.1).

**Linear Projection Layer** The first layer projects individual points $\mathbf{x}_{i,t} \in \mathbb{R}^C$ to vectors $\mathbf{u}_{i,t} \in \mathbb{R}^F$ with a fixed number of channels $F$. Random timestamp masking is applied to each $\mathbf{u}_i$ independently after the linear projection (only during training), as part of the *contextual consistency* framework (Yue et al., 2022).

**Time-Embedding Module** The time-embedding module $h_\psi$ is responsible for learning time-related features $\boldsymbol{\tau}_t$ (trend, periodicity, distribution shifts etc.) directly from the time series sample indices $t$. The time-embedding function is not fixed like a transformer's positional encoding module, it is learned jointly with the rest of the encoder. This makes the time-embeddings flexible, they adapt to the data at hand. The choice of architecture for the time-embedding module can impact performance in downstream tasks, and is discussed in Appendix A.4. For general applications, we recommend using Time2Vec (Kazemi et al., 2019), which captures trend and periodicity. To the best of our knowledge, T-Rep is the first model to combine a time-embedding module and a convolutional encoder in self-supervised learning for time series.

The time-embedding module must return vectors which define a probability distribution (positive components that sum to 1). This is due to the use of statistical divergence measures in a pretext task, which is detailed in section 4.2.1. We find experimentally that the optimal way to satisfy this

constraint is by applying a sigmoid activation to the final layer of the module, and then dividing each element by the vector sum:

$$(\boldsymbol{\tau}_t)_k = \frac{\sigma(h_\psi(t))_k}{\sum_{j=1}^{K} \sigma(h_\psi(t))_j} \,, \tag{3}$$

where $\boldsymbol{\tau}_t$ contains $K$ elements, $\sigma(\cdot)$ is the sigmoid function and $h_\psi$ is the time-embedding module parameterised by $\psi$. Time-embeddings $\boldsymbol{\tau}_t$ are concatenated with vectors $\mathbf{u}_{i,t}$ after the linear projection, and the vectors $[\mathbf{u}_{i,t} \ \boldsymbol{\tau}_t]^T$ are fed to the encoder $f_\theta$.

**Temporal Convolution Network (TCN) Encoder** The main body of the encoder, $f_\theta(\cdot)$, is structured as a sequence of residual blocks, each containing two layers of 1D dilated convolutions interleaved with GeLU activations. The convolution dilation parameter increases with the network depth, to first focus on local features and then longer-term dependencies: $d = 2^i$, where $i$ is the block index. Dilated convolutions have proven to be very effective in both supervised and unsupervised learning for time series (Zhou et al., 2021; Tonekaboni et al., 2021; Bai et al., 2018).

## 4.2 PRETEXT TASKS

We present below two novel SSL pretext tasks which leverage time-embeddings, and are designed to complement each other. The first, 'Time-embedding Divergence Prediction', describes *how* the information gained through time-embeddings should structure the latent space and be included in the time series representations. On the other hand, the 'Time-embedding-conditioned Forecasting' task focuses on *what* information the time-embeddings and representations should contain.

### 4.2.1 TIME-EMBEDDING DIVERGENCE PREDICTION

The first pretext task developed aims to integrate the notion of time in the latent space structure. It consists in predicting a divergence measure between two time-embeddings $\boldsymbol{\tau}$ and $\boldsymbol{\tau}'$, given the representations at the corresponding time steps. The purpose of this task is for distances in the latent space to correlate with temporal distances, resulting in smoother latent trajectories than with contrastive learning.

Let us define this regression task formally. Take a batch $X \in \mathbb{R}^{B \times T \times C}$, from which we sample $\mathbf{x}_{i,t}$ and $\mathbf{x}_{j,t'}$ $\forall i, j \in [0, B]$ and $t, t' \in [0, T]$ s.t. $t \neq t'$. The task input is the difference $\mathbf{z}_{i,t} - \mathbf{z}'_{j,t'}$, where $\mathbf{z}_{i,t}$ is the representation of $\mathbf{x}_{i,t}$ under the context $c$ and $\mathbf{z}'_{j,t'}$ is the representation of $\mathbf{x}_{j,t'}$ under the context $c'$. Taking representations under different contexts further encourages context-invariant representations, as detailed in section 3.2. The regression target is $y = \mathcal{D}(\boldsymbol{\tau}, \boldsymbol{\tau}')$. $\boldsymbol{\tau}$ and $\boldsymbol{\tau}'$ are the respective time-embeddings of $t$ and $t'$, and $\mathcal{D}$ is a measure of statistical divergence, used to measure the discrepancy between the time-embedding distributions. We use the Jensen-Shannon divergence (JSD), a smoothed and symmetric version of the KL divergence (Lin, 1991). The task loss is:

$$\mathcal{L}_{div} = \frac{1}{M} \sum_{(i,j,t,t') \in \Omega}^{M} \left( \mathcal{G}_1 \left( \mathbf{z}_{i,t} - \mathbf{z}'_{j,t'} \right) - JSD(\boldsymbol{\tau}_t \,||\, \boldsymbol{\tau}_{t'}) \right)^2 \,, \tag{4}$$

where $\Omega$ is the set (of size $M$) of time/instance indices for the randomly sampled pairs of representations. Using divergences allows us to capture *how* two distributions measures differ, whereas a simple norm could only capture by *how much* two vectors differ. This nuance is important - suppose the time-embedding is a 3-dimensional vector that learned a hierarchical representation of time (equivalent to seconds, minutes, hours). A difference of 1.0 on all time scales (`01:01:01`) represents a very different situation to a difference of 3.0 hours and no difference in minutes and seconds (`03:00:00`), but could not be captured by a simple vector norm.

### 4.2.2 TIME-EMBEDDING-CONDITIONED FORECASTING

Our second pretext task aims to incorporate predictive information in the time-embedding vectors, as well as context-awareness in the representations, to encourage robustness to missing data. The task takes in the representation of a time series at a specific timestep, and tries to predict the representation vector of a nearby point, conditioned on the target's time-embedding.

The input used is the concatenation $[\mathbf{z}_{i,t} \; \boldsymbol{\tau}_{t+\Delta}]^T$ of the representation $\mathbf{z}_{i,t} \in \mathbb{R}^F$ at time $t$ and the time-embedding of the target $\boldsymbol{\tau}_{t+\Delta} \in \mathbb{R}^K$. $\Delta_{max}$ is a hyperparameter to fix the range in which the prediction target can be sampled. The target is the encoded representation $\mathbf{z}_{i,t+\Delta}$ at a uniformly sampled timestep $t + \Delta$, $\Delta \sim \mathcal{U}[-\Delta_{max}, \Delta_{max}]$. The input is forwarded through the task head $\mathcal{G}_2 : \mathbb{R}^{F+K} \mapsto \mathbb{R}^F$, a 2-layer MLP with ReLU activations. The loss is a simple MSE given by:

$$\mathcal{L}_{pred} = \frac{1}{MT} \sum_{j \in \Omega_N}^M \sum_{t \in \Omega_T}^T \left( \mathcal{G}_2 \left( \left[ \mathbf{z}_{i,t}^{(c_1)} \; \boldsymbol{\tau}_{t+\Delta_j} \right]^T \right) - \mathbf{z}_{i,t+\Delta_j}^{(c_2)} \right)^2 , \tag{5}$$

where $\Delta_j \sim \mathcal{U}[-\Delta_{max}, \Delta_{max}]$, $\Omega_M$ and $\Omega_T$ are the sets of randomly sampled instances and timesteps for each batch, whose respective cardinalities are controlled by hyperparameters $M$ and $T$. The contexts $c_1$ and $c_2$ are chosen randomly from $\{c, c'\}$, so they may be identical or different, further encouraging contextual consistency. Conditioning this prediction task on the time-embedding of the target forces the model to extract as much information about the signal as possible from its position in time. This results in more information-dense time-embeddings, which can be leveraged when working with individual trajectories for forecasting and anomaly detection.

In practice, we choose a short prediction range $\Delta_{max} \leqslant 20$, as the focus is not to build representations tailored to forecasting but rather 'context-aware' representations. This context-awareness is enforced by making predictions backwards as well as forwards, encouraging representations to contain information about their surroundings, making them robust to missing timesteps. Longer prediction horizons would push representations to contain more predictive features than spatial features, biasing the model away from use-cases around classification, clustering and other 'comparative' or instance-level downstream tasks.

A key objective of this pretext task is to build resilience to missing data. This is done by (1) learning information-dense time-embeddings, which are available even when data is missing, and (2) by learning context-aware representations, which can predict missing timesteps in their close vicinity.

## 5 EXPERIMENTS

This sections presents the experiments conducted to evaluate T-Rep's learned representations. Because of the variety of downstream tasks, we perform no hyperparameter tuning, and use the same hyperparameters across tasks. Further, the same architectures and hyperparameters are used across all evaluated models where possible, to ensure a fair comparison. Experimental details and guidelines for reproduction are included in Appendix A.2 and A.3. The code written to produce these experiments has been made publicly available[1].

### 5.1 TIME SERIES ANOMALY DETECTION

We perform two experiments, point-based anomaly detection on the Yahoo dataset (Nikolay Laptev, 2015), and segment-based anomaly detection on the 2019 PhysioNet Challenge's Sepsis dataset (Reyna et al., 2020a; Goldberger et al., 2000). We chose to include both types of tasks as segment-based anomaly detection tasks help avoid the bias associated with point-adjusted anomaly detection (Kim et al., 2022). Yahoo contains 367 synthetic and real univariate time series, featuring outlier and change point anomalies. Sepsis is a real-world dataset containing multivariate time series from $40,336$ patients in intensive care units,

|  | Yahoo (F1) | Sepsis (F1) |
|---|---|---|
| Baseline | 0.110 | 0.241 |
| TS2Vec | 0.733 | 0.619 |
| **T-Rep** (Ours) | **0.757** | **0.666** |

Table 1: Time series anomaly detection F1 scores, on the Yahoo dataset for point-based anomalies and Sepsis datasets for segment-based anomalies. Anomalies include outliers as well as change-points. TS2Vec results are reproduced using official source code (Zhihan Yue, 2021).

featuring noisy and missing data. The task consists in detecting sepsis, a medical anomaly present in just 2.2% of patients. On both datasets, we compare T-Rep to the self-supervised model TS2Vec (Yue et al., 2022), as well as a baseline following the same anomaly detection protocol as TS2Vec

[1]https://github.com/Let-it-Care/T-Rep

and T-Rep on each dataset, but using the raw data. We follow a streaming anomaly detection procedure (Ren et al., 2019) on both datasets. Details on the procedures can be found in Appendix A.2.3, which also details the pre-processing applied to Sepsis.

Table 7 shows the evaluation results (more detailed results are presented in Appendix A.9). T-Rep achieves the strongest performance in both datasets, with an F1 score of 75.5% on Yahoo and 66.6% on Sepsis. This respectively represents a 2.4% and 4.8% increase on the previous SOTA TS2Vec (Yue et al., 2022). T-Rep's performance can be attributed to its detailed understanding of temporal features, which help it better detect out-of-distribution or anomalous behaviour. It achieves this performance with a latent space dimension of 200 on Yahoo, which is smaller than the 320 dimensions used by TS2Vec (Yue et al., 2022), further showing that T-Rep learns more information-dense representations than its predecessors.

## 5.2 TIME SERIES CLASSIFICATION

The classification procedure is similar to that introduced by Franceschi et al. (2019): a representation $z$ is produced by the self-supervised model, and an SVM classifier with RBF kernel is then trained to classify the representations (see Appendix A.2 and A.3 for procedure and reproduction details). We compare T-Rep to SOTA self-supervised models for time series: TS2Vec (Yue et al., 2022), T-Loss (Franceschi et al., 2019), TS-TCC (Eldele et al., 2021), TNC (Tonekaboni et al., 2021), Minirocket Dempster et al. (2021) and a DTW-based classifier (Müller, 2007). These models are evaluated on the UEA classification archive's 30 multivariate time series, coming from diverse domains such as medicine, sensor systems, speech, and activity recognition Dau et al. (2019).

| Method | 30 UEA datasets | |
| --- | --- | --- |
| | Avg. Acc. | Avg. Difference (%) |
| T-Loss | 0.657 | 33.6 |
| TS2Vec | 0.699 | 2.1 |
| TNC | 0.671 | 12.3 |
| TS-TCC | 0.667 | 10.0 |
| DTW | 0.650 | 43.6 |
| Minirocket | **0.707** | 4.8 |
| **T-Rep** (Ours) | 0.706 | – |

Table 2: Multivariate time series classification results on the UEA archive. DTW, TNC, and TS-TCC results are taken directly from Yue et al. (2022), while TS2Vec and Minirocket results are reproduced using the official code. 'Avg. Difference' measures the relative difference in accuracy brought by T-Rep compared to a given model.

Evaluation results are summarised in Table 2, and full results are shown in Appendix A.7, along with more details on the chosen evaluation metrics. Table 2 shows that T-Rep has an accuracy $+2.1\%$ higher than TS2Vec and $+4.8\%$ higher than Minirocket on average. In terms of average accuracy, T-Rep's accuracy outperforms all competitors except Minirocket, which has a 0.001 lead. TS2Vec's performance is very close to T-Rep's, only 0.07 lower. T-Rep has a positive 'Avg. difference' to Minirocket despite having a lower 'Avg. Acc.' because Minirocket often performs slightly better than T-Rep, but when T-Rep is more accurate, it is so by a larger margin. It is important to note that Minirocket was developed specifically for time series classification (and is the SOTA as of this date), while T-Rep is a general representation learning model, highlighting its strengths across applications. The latent space dimensionality is set to 320 for all baselines, except Minirocket which uses 9996 dimensions (as per the official code). T-Rep uses only 128 dimensions, but leverages the temporal dimension of its representations (see Appendix A.2.4 for details).

## 5.3 TIME SERIES FORECASTING

We perform a multivariate forecasting task on the four public ETT datasets (ETTh1, ETTh2, ETTm1, ETTm2), which contain electricity and power load data, as well as oil temperature (Zhou et al., 2021). Forecasting is performed over multiple horizons, using a procedure described in Appendix A.2.2. We evaluate T-Rep against TS2Vec (Yue et al., 2022), a SOTA self-supervised model for time series, but also Informer (Zhou et al., 2021), the SOTA in time series forecasting, as well as TCN (Bai et al., 2018), a supervised model with the same backbone architecture as T-Rep, and a linear model trained on raw data. Aggregate results, averaged over all datasets and prediction horizons, are presented in Table 3 (full results are in Appendix A.8).

Table 3 shows that T-Rep achieves the best average scores, in terms of both MSE and MAE, with a 24.2% decrease in MSE on the supervised SOTA Informer (Zhou et al., 2021), and a slight improvement of 1.80% on the self-supervised SOTA TS2Vec (Yue et al., 2022). It also achieves a better average rank, ranking first more than any other model. Furthermore, the linear baseline is the second model that most frequently ranks first, beating TS2Vec in this metric. However, this model does not perform well in all datasets and therefore ranks 3rd in terms of MAE and MSE. Interestingly, most existing self-supervised methods for time series use high-dimensional latent spaces ($F = 320$ dimensions per timestep) (Franceschi et al., 2019; Tonekaboni et al., 2021), which is thus used to produce baseline results in Table 3. This can be an issue for downstream applications, which might face the *curse of dimensionality* (Verleysen & François, 2005). T-Rep, however, outperforms all baselines with a latent space that is almost 3 times smaller, using only $F = 128$ dimensions. T-Rep's superior performance can be attributed to its comprehensive treatment of time, capturing trend, seasonality, or distribution shifts of more easily with its time-embedding module.

| | **T-Rep** (Ours) | | TS2Vec | | Informer | | TCN | | Linear Baseline | |
|---|---|---|---|---|---|---|---|---|---|---|
| | MSE | MAE | MSE | MAE | MSE | MAE | MSE | MAE | MSE | MAE |
| Avg. Rank | **1.90** | **1.85** | 2.40 | 2.45 | 3.3 | 3.55 | 4.35 | 4.1 | 3.05 | 3.05 |
| Ranks $1^{st}$ | **8** | **8** | 2 | 1 | 3 | 3 | 1 | 1 | 6 | 7 |
| Avg. | **0.986** | **0.702** | 1.004 | 0.712 | 1.300 | 0.820 | 2.012 | 1.205 | 1.017 | 0.727 |

Table 3: Multivariate time series forecasting results on the ETT datasets, over 5 different prediction horizons. The presented results are averaged over all datasets/prediction horizons. The 'Ranks $1^{st}$' metric counts the number of times a model ranks $1^{st}$ amongst its competitors. Results for all models are based on our own reproductions, using the official code for each model (see Appendix A.2).

## 5.4 ROBUSTNESS TO MISSING DATA

T-Rep was developed with medical and HAR applications in mind, requiring strong resilience to missing data, which we evaluate in two different experiments. In the first *qualitative* experiment, we visualize representations of incomplete time series using the DodgerLoopGame dataset from the UCR archive Dau et al. (2019), which features a continuous segment of 25 missing timesteps (see top row of Figure 2.a). We then visualised a heatmap of T-Rep and TS2Vec's time series representations (bottom row), showing the 15 dimensions with highest variance. T-Loss (Franceschi et al., 2019) is not included as it does not produce timestep-level representations. For TS2Vec, the representations of missing timesteps very much stand out (they are brighter, reflecting higher values), illustrating that the model is struggling to interpolate and instead produces out-of-distribution representations. On the other hand, T-Rep produces much more plausible representations for these missing timesteps with smoother transitions in and out of the area with missing points, as well as realistic interpolations, matching the data distribution of the surrounding area.

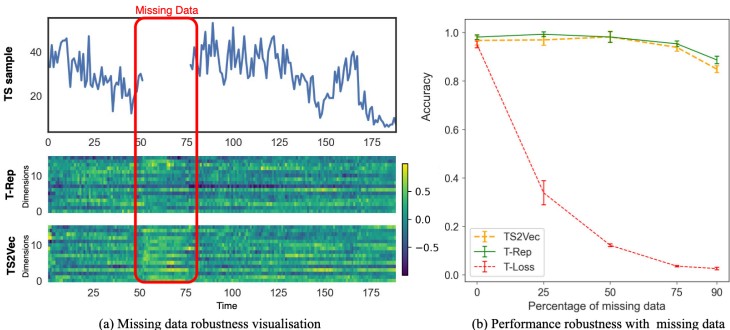

(a) Missing data robustness visualisation        (b) Performance robustness with missing data

Figure 2: Illustration of T-Rep's robustness to missing data on UCR archive datasets. (a) shows heatmap representations of T-Rep and TS2Vec when faced with missing data, and (b) shows accuracy against percentage of missing data in a classification task for T-Rep, TS2Vec (Yue et al., 2022) and T-Loss (Franceschi et al., 2019). Error bars denote the standard deviation over 6 train-test runs.

Secondly, we decided to perform a more *quantitative* experiment, examining classification accuracy for different amounts of missing data, on the ArticularyWordRecognition dataset of the UCR archive (Dau et al., 2019). We compare T-Rep's performance to TS2Vec (Yue et al., 2022) and T-Loss, a self-supervised representation learning model for time series, specifically designed for downstream classification and clustering tasks (Franceschi et al., 2019). The results are unequivoqual, T-Rep's performance is extremely resilient to missing data, starting at 98% accuracy with the complete dataset, and dropping by only 1.3% when faced with 75% missing data, and finally reaching 86.5% with only 10% of the available data (green curve of Figure 2.b). The performance of TS2Vec is also very strong (orange curve), following a similar trend to T-Rep with 2% less accuracy on average, and a more pronounced dip in performance when 90% of the data is missing, dropping to 82.7%. On the other hand, T-Loss is much more sensitive to any missing data. Its performance decreases exponentially to reach 2.6% when presented with 90% missing data (red curve).

## 5.5 ABLATION STUDY

To empirically validate T-Rep's components and pretext tasks, we conduct ablation studies on forecasting and anomaly detection tasks using the ETT datasets for forecasting and the PhysioNet Challenge's Sepsis dataset for anomaly detection. Unless specified, Time2Vec is the chosen time-embedding method. **w/o TE-conditioned forecasting** assigns a weight of 0.0 to the 'time-embedding-conditioned' forecasting task, redistributing weights evenly. **w/o TE divergence prediction** behaves similarly, but for the 'time-embedding divergence prediction' task. **w/o New pretext tasks** retains only the time-embedding module and the two TS2Vec pretext tasks, isolating the impact on performance of different time-embedding architectures. We explore a fully-connected two-layer MLP with ReLU activations (**w/ MLP TE module**) and a vector of RBF features (**w/ Radial Basis Features TE module**). The original TS2Vec model (**w/o TE module**) is also included in the ablation study, lacking any of T-Rep elements.

|  | Forecasting | Anomaly Detection |
|---|---|---|
| **T-Rep** | **0.986** | **0.665** |
| *Pretext tasks* | | |
| w/o TE-conditioned forecasting | 1.022 (+3.7%) | 0.392 (-41%) |
| w/o TE divergence prediction | 1.003 (+1.7%) | 0.634 (-4.7%) |
| w/o New pretext tasks | 0.999 (+1.3%) | 0.513 (-22.8%) |
| *Architecture* | | |
| w/ MLP TE module | 1.008 (+2.2%) | 0.443 (-33.3%) |
| w/ Radial Basis Features TE module | 1.007 (+2.1%) | 0.401 (-39.7%) |
| w/o TE module (=TS2Vec) | 1.004 (+1.8%) | 0.610 (-8.2%) |

Table 4: Ablation study results on ETT forecasting datasets (measured in MSE) and the Sepsis anomaly detection dataset (measured in F1 score). Percentage changes are calculated as the relative difference between a modified model's performance and T-Rep's.

Results in Table 4 confirm that the proposed pretext tasks and the addition of a time-embedding module to the encoder contribute to T-Rep's performance: removing any of these decreases the scores in both tasks. These results also illustrate the interdependency of both tasks, as in forecasting, only leaving one of the tasks obtains worse results than removing both pretext tasks. It also justifies our preferred choice of time-embedding, since Time2Vec (Kazemi et al., 2019) outperforms the other 2 architectures in both tasks.

## 6 CONCLUSION

We present T-Rep, a self-supervised method for learning representations of time series at a timestep granularity. T-Rep learns vector embeddings of time alongside its encoder, to extract temporal features such as trend, periodicity, distribution shifts etc. This, alongside pretext tasks which leverage the time-embeddings, allows our model to learn detailed temporal dependencies and capture any periodic or irregularly recurring patterns in the data. We evaluate T-Rep on classification, forecasting, and anomaly detection tasks, where it outperforms existing methods. This highlights the ability of time-embeddings to capture temporal dependencies within time series. Further, we demonstrate T-Rep's efficiency in missing data regimes, and provide visualisation experiments of the learned embedding space, to highlight the interpretability of our method.

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

## A APPENDIX

### A.1 LATENT SPACE STRUCTURE

This section aims to assess the representations produced by T-Rep in different settings, and show their interpretability. We perform 2 experiments, the to examine the intra-instance latent space structure as well as the inter-instance structure.

For the first experiment, we choose 2 datasets from the UCR archive (Dau et al., 2019) and create one synthetic dataset. We then plot a time series sample for each dataset and show a heatmap of the corresponding T-Rep representation evolving through time (only the 15 dimensions with highest variance are shown for each representation vector).

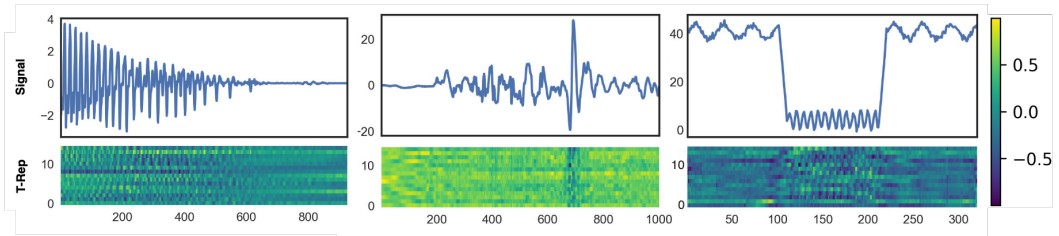

Figure 3: Time series representation trajectories in the latent space. The top row shows the input signal, and the bottom row shows a heatmap of the representations through time. Only the 15 most varying dimensions are shown for each representation. Data for the two left-most figures comes from UCR archive datasets, and the righ-most figure's data is synthetic.

Figure 3 showcases a variety of input signal properties captured by T-Rep. The leftmost plot showcases a sample from the Phoneme dataset (Dau et al., 2019), and we see that periodicity is captured by T-Rep, in particular the change in frequency and decrease in amplitude. Plot (b) is an extract of the Pig Central Venous Pressure medical dataset (Dau et al., 2019), where T-Rep accurately detects an anomaly in the input signal, represented by a sharp change in value (illustrated by a color shift from green to blue) at the corresponding spike. Finally, the rightmost plot showcases a synthetic dataset with an abrupt distribution shift. T-Rep's representation captures this with lighter colors and a change in frequency, as in the original signal. Finally, we can observe how T-Rep produces coherent representations for missing data in Figure 2.a (see section 5.4 for details). This experiment highlights the interpretability of the representations learned by T-Rep, accurately reflecting properties of the original signal.

To study the global latent space structure (inter-instance), we look at UMAP plots (McInnes et al., 2018) of representations learned by T-Rep on five UCR datasets Dau et al. (2019). Each instance is represented by a point, colored by the class it belongs to, to show whether the latent space structure discriminates the datasets' different classes. We include representations learned by TS2Vec to see how the new components of T-Rep affect the learned latent space. Results are presented in Figure 4.

Looking at the UMAPs across the 128 UCR datasets, we noticed that the latent space structure of T-Rep and TS2Vec is often similar, and/or of similar quality. This is shown in the two rightmost plots of Figure 4, and can be explained by the fact that T-Rep focuses on learning temporal features (and uses TS2Vec's tasks for spatial features), resulting in stronger differences in the evolution of representations through time (see Figure 3).

To focus on learning detailed temporal features, T-Rep puts a lower weight on TS2Vec's spatial-features pretext tasks. While one might intuitively think this results in lower quality spatial features, this is not what our experiments show. Indeed, for none of the tested datasets did we observe a significant downgrade in latent space structure, which is further supported by the classification results presented in section 5.2, where T-Rep outperforms TS2Vec by a small margin. These results show that learning a detailed temporal structure within an instance constitutes a useful feature to

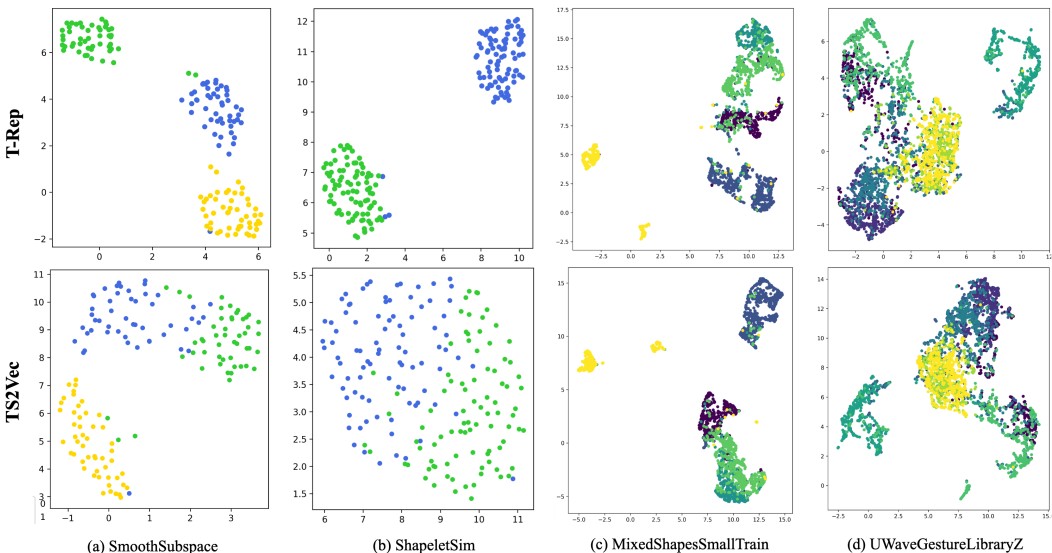

Figure 4: UMAP visualisations of the learned representations of T-Rep and TS2Vec on 5 UCR datasets. Each color represents a different class. TS2Vec results are produced using the official code (Zhihan Yue, 2021).

compare various instances. This is illustrated by the two leftmost plots of Figure 4, where we see much more cohesive and separated clusters for T-Rep than TS2Vec. This experiment truly highlights the versatile nature of T-Rep, improving the quality of representations at both the dataset scale (inter-instance discrimination) and individual time series (timestep granularity).

## A.2 EXPERIMENTAL DETAILS

The implementation of the models is done in Python, using Pytorch 1.13 (Paszke et al., 2019) for deep learning and scikit-learn (Pedregosa et al., 2011) for SVMs, linear regressions, pre-processing etc. We use the Adam optimizer (Kingma & Ba, 2014) throughout, and train all models on a single NVIDIA GeForce RTX 3060 GPU with Cuda 11.7.

As mentioned in the Experiments section, no hyperparameter tuning was performed. We use a batch size of 16, a learning rate of 0.001, set the maximum number of epochs to 200 across all datasets and tasks. We use 10 residual blocks for the encoder network, and set hidden channel widths to 128. The kernel size of all convolution layers is set to 3. The weighting of pretext tasks is detail in section A.2.1 below. As for the choice of time-embedding, we use a fully-connected two-layer MLP with ReLU activations for the classification tasks, and Time2Vec for the forecasting and anomaly detection tasks (see Appendix A.4 for additional information). For all experiments (unless stated otherwise), models are trained and tested 10 times per dataset to mitigate stochasticity. No random seed is used in any experiments.

### A.2.1 PRETEXT TASK WEIGHTS

As shown in Figure 1, the loss passed to hierarchical loss computation framework is a linear combination of pretext task's loss:

$$\mathcal{L} = \sum_{i=1}^{4} \alpha_i \cdot \ell_i \qquad \text{s.t.} \quad \sum_{i=1}^{4} \alpha_i = 1 \,,$$

where $\alpha_i$ designates task $i$'s weight. Choosing the task weights can be seen as a semantic as well as an optimisation problem: the extracted features will of course depend on the weights attributed to each task, but the choice of weights will also impact the optimisation landscape of the final

loss. Balancing these two factors is a complex and interesting research direction, which we leave as future work. For all pretext tasks (instance-wise contrasting, temporal contrasting, time-embedding divergence prediction, time-embedding-conditioned forecasting), we use the same weights: $\alpha_i = 0.25 \; \forall i \in [1, 4]$. This is the case across all downstream tasks, no task-specific tuning is performed.

### A.2.2 FORECASTING

The multivariate forecasting task consists in taking the $L$ previous points $X = \{\mathbf{x}_{t-L+1}, ..., \mathbf{x}_{t-1}, \mathbf{x}_t\} \in \mathbb{R}^{L \times C}$ and using them to predict the next $H$ points $y = \{\mathbf{x}_{t+1}, ..., \mathbf{x}_{t+H}\} \in \mathbb{R}^{H \times C}$, where $H$ is the prediction horizon. Such $X$ and $y$ are used to train the supervised fore-casting baselines. For self-supervised models, the forecasting model is a simple ridge regression, taking as input $\mathbf{z}_t$ (the representation of $\mathbf{x}_t$), and trying to predict $y$. The input time series are normalised using the z-score, each channel being treated independently. We use no covariates such as minute, hour, day-of-the-week etc., to avoid interference with the temporal features learned by the time-embeddings. To accurately evaluate the benefits of time-embeddings, we wanted to ensure the model would not rely on other temporal features (the covariates).

The forecasting model is a ridge regression (linear regression with an $L_2$ regularisation term, weighted by a parameter $\alpha$) trained directly on the extracted representations. $\alpha$ is chosen using grid search on a validation set, amongst the following values: $\alpha \in \{0.1, 0.2, 0.5, 1, 2, 5, 10, 20, 50, 100, 500, 1000\}$.

The train/validation/test split for the ETT datasets is 12/4/4 months. The prediction horizon $H$ ranges from 1 to 30 days for the hourly data (ETTh$_1$ and ETTh$_2$), and from 6 hours to 7 days for minute data (ETTm$_1$ and ETTm$_2$). The evaluation metrics, MSE and MAE, are respectively given by:

$$\text{MSE} = \frac{1}{HB} \sum_{i=1}^{H} \sum_{j=1}^{B} (x_{t+i}^{j} - \hat{x}_{t+i}^{j})^2 , \tag{6}$$

$$\text{MAE} = \frac{1}{HB} \sum_{i=1}^{H} \sum_{j=1}^{B} |x_{t+i}^{j} - \hat{x}_{t+i}^{j}| , \tag{7}$$

where $x_{t+i}^{j}$ is the true value, $\hat{x}_{t+i}^{j}$ is the prediction, $N$ is the number of time series instances indexed by $j$. The reported MSE and MAE are thus averaged over all prediction horizons and instances. To mitigate the effects of stochasticity during training, each model is trained and tested 10 times on each dataset and prediction horizon, and the reported results are the mean.

### A.2.3 ANOMALY DETECTION

**Yahoo**

The Yahoo anomaly detection task follows a streaming evaluation procedure (Ren et al., 2019), where given a time series segment $\{x_1, ..., x_t\}$, the goal is to infer if the last point $x_t$ is anomalous. To avoid drifting, the input data is differenced $d$ times, following the number of roots $d$ found by an ADF test (Dickey & Fuller, 1979). For fair comparison, we follow the anomaly detection procedure of Yue et al. (2022), which computes an *anomaly score* $\alpha_t$ for $x_t$. $\alpha_t$ is given by the dissimilarity between representations where the observation of interest $x_t$ has been masked vs. unmasked. Since the model was trained with random masking, it should output a coherent representation even with a masked input, resulting in a small $\alpha_t$. However, if the point $x_t$ presents an anomaly, the masked representation will be very different from the unmasked representation, as it anticipated 'normal' behaviour, not an anomaly, resulting in a high value of $\alpha_t$. More precisely, $\alpha_t$ is given by:

$$\alpha_t = ||r_t^u - r_t^m||_1 , \tag{8}$$

where $r_t^u$ is the representation of the unmasked $x_t$ and $r_t^m$ is the representation of the masked observation. The anomaly score is adjusted by the average of the preceding $Z$ points:

$$\alpha_t^{adj} = \frac{\alpha_t - \bar{\alpha}_t}{\bar{\alpha}_t} , \tag{9}$$

where $\bar{\alpha}_t = \frac{1}{Z}\sum_{i=t-Z}^{Z}\alpha_i$. At inference time, anomalies are detected if the adjusted anomaly score $\alpha_t^{adj}$ is above a certain threshold: $\alpha_t^{adj} > \mu + \beta\cdot\sigma$, where $\mu$ and $\sigma$ are the mean and standard deviation of the historical scores, and $\beta$ is a hyperparameter. If an anomaly is detected within a delay of 7 timesteps from when it occured, it is considered correct, following the procedure introduced in Ren et al. (2019). To mitigate the effects of stochasticity during training, each model is trained and tested 10 times, and the reported results are the mean.

**Sepsis**

The anomaly detection procedure followed for Sepsis is different from Yahoo's. We use a supervised method, since there is only one type of anomaly (sepsis) to detect, so we can fit a model to it, obtaining higher scores than a purely unsupervised method (as is done for Yahoo). One could not do that with a dataset like Yahoo, where there are various anomalies which cannot be explicitly classified. Despite being solved in a supervised manner, we believe sepsis detection remains an *anomaly detection* task and is differs from *classification* because we are classifying individual timesteps rather than entire time series, thus evaluating different characteristics of T-Rep. To perform well, T-Rep must thus produce representations with accurate *intra-instance* structure, where healthy timesteps are very different from abnormal (sepsis) timesteps (which is learned in a *self-supervised* manner) . This is further supported by the very low number of positive samples (2.2%), so training a classification model on raw data is not efficient. For these reasons, although we used a supervised model on top of our representation, we consider this task to be an *anomaly detection* task.

The Sepsis dataset is firstly pre-processed by forward-filling all missing values and features which have lower granularity than the pysiological data, to ensure we have data at all timesteps for all features. The length of time-series is then standardised to 45 (patient ICU stay length varies in the raw data), corresponding to the last 45 hours of each patient in the ICU. We then subselect ten features based on medical relevance ('HR', 'Temp', 'Resp', 'MAP', 'Creatinine', 'Bilirubin direct', 'Glucose', 'Lactate', 'Age', 'ICULOS'). We have found this to be rather 'light' pre-processing compared to what top-performers in the 2019 PhysioNet Challenge did. This is done on purpose, to further highlight gaps between model performances.

We obtain a dataset $\mathbf{X} \in \mathbb{R}^{N \times 45 \times 10}$. We train self-supervised models TS2Vec and T-Rep on this dataset, and then encode individual time series in sliding windows of length 6 (this window length seemed like a medically sensible lookback period, but we performed no parameter tuning). We thus obtain latent representations

$$\mathbf{z}_{t-6:t} = f_\theta(\mathbf{x}_{t-6:t}) \tag{10}$$

where $f_\theta$ is the self-supervised model (T-Rep or TS2Vec) and $\mathbf{z}_{t-6:t} \in \mathbb{R}^{6 \times F}$, with latent dimensionality $F = 32$. A very small latent space is used since we pre-selected 32 features. Given $\mathbf{z}_{t-6:t}$, we try to predict whether $\mathbf{z}_t$ showcases sepsis, which is framed as a binary classification problem. This classification task is then solved using an SVM classifier with RBF kernel is trained using the representations. The penalty hyperparameter $C$ is chosen through cross-validation, in the range $\{10^k | k \in [\![-4, 4]\!]\}$. The linear baseline uses the same method, but using the raw data $\mathbf{x}_{t-6:t}$ instead of learned representations $\mathbf{z}_{t-6:t}$.

### A.2.4 CLASSIFICATION

The classification procedure for TS-TCC, TNC, and T-Loss is the same as in Franceschi et al. (2019). Instance-level representations are produced by the self-supervised model, and an SVM classifier with RBF kernel is trained using the representations. The penalty hyperparameter $C$ is chosen through cross-validation, in the range $\{10^k | k \in [\![-4, 4]\!]\}$.

For T-Rep and TS2Vec, the procedure is slightly different, leveraging the fact that the temporal granularity of representations is controllable by the user (from timestep-level to instance-wide), thanks to the *hierarchical loss* framework Yue et al. (2022). We use this to produce representations which maintain a temporal dimension, whose length is controlled by hyperparameter $W$: the latent vectors $z$ have dimension $\mathbb{R}^{\lfloor \frac{T}{W} \rfloor \times F}$. This is achieved by applying a maxpool operation with kernel size $k = \lfloor \frac{T}{W} \rfloor$ and stride $s = \lfloor \frac{T}{W} \rfloor$ to the timestep-level representation $z \in \mathbb{R}^{T \times F}$. The

representation $z \in \mathbb{R}^{\lfloor \frac{T}{W} \rfloor \times F}$ is then flattened to obtain $z_{flat} \in \mathbb{R}^{\lfloor \frac{T}{W} \rfloor \cdot F}$, which is fed to an SVM with RBF kernel for classification, following the exact same protocol as in Franceschi et al. (2019). We refer to this new procedure as *TimeDim* in Table 5.

Maintaining the temporal dimension in the representations is advantageous, as it gives a better snapshot of a time series' latent trajectory, compared to simply using one instance-wide vector. This is especially important for T-Rep, whose innovation lies in the treatment of time and its incorporation in latent trajectories. Because keeping some temporality increases the dimensionality of the vector $z_{flat}$ passed to the SVM, we use a smaller latent dimension $F$ to compensate. In practice we use $W = 10$ and $F = 128$, but no hyperparameter tuning was performed, so these may not be optimal parameters. As in all other experiments, the reported results are the mean obtained after training and testing each model 10 times per dataset.

Interestingly, for TS2Vec, the *TimeDim* procedure does not result in a performance gain, so the scores obtained with the original procedure from Franceschi et al. (2019) (using instance-wide representations) are reported in Table 2. To summarise, for the results reported in Table 2 (in the paper's main body), T-Rep is the only model to use the *TimeDim* procedure. TS2Vec, TNC, T-Loss, TS-TCC all use the procedure introduced by Franceschi et al. (2019). TS2Vec results for both procedures are reported in Table 5, which contains scores on individual datasets, as well as aggregate metrics.

### A.3    REPRODUCTION DETAILS FOR BASELINES

**TS2Vec:** The results for TS2Vec (Yue et al., 2022) have been reproduced using the official code (found here). The hyperparameters used for TS2Vec are exactly those of the original paper (except for the batch size, set to 16 in our reproduction), and the same as T-Rep where possible. The only notable difference in hyperparameters is that T-Rep uses slightly wider hidden channels to accommodate the additional use of a time-embedding vector. The results published in the TS2Vec paper weren't used for the forecasting task because of slight changes we made to the experimental setup, specifically removing the use of covariates in forecasting. For anomaly detection and classification, we used our own reproduction because the results slightly differed from the originally published results, which we attribute to the following reasons. Firstly, the change in batch size from 8 to 16 for us, which the original paper says to be important (Yue et al., 2022). Secondly, we introduced a new classification procedure, so we had to test TS2Vec in this new setup. Further, we train and test all models 10 times to mitigate stochasticity and obtain more stable results, which is not the case for the results of the original TS2Vec paper (Yue et al., 2022). Also, we do not use any random seed, while the TS2Vec codebase Zhihan Yue (2021) shows a single run with a random seed was used to produce their results.

**Informer:** For the ETTh$_1$, ETTh$_2$ and ETTm$_1$ datasets, we use the results published in the Informer paper (Zhou et al., 2021). However, the paper doesn't include results on ETTm$_2$, so we obtain these results using the official code (found here) and the default hyperparameters (which are also used in the original paper).

**TCN:** T-Rep's encoder architecture is a TCN network (Bai et al., 2018), so we reused our code to implement the forecasting TCN model, changing only the training part to use a supervised MSE loss. The hyperparameters (channel widths, learning rate, optimiser) are the same as for T-Rep.

For the classification tasks, the results of TS-TCC (Eldele et al., 2021), TNC (Tonekaboni et al., 2021), and DTW (Müller, 2007) are taken from the TS2Vec paper (Yue et al., 2022), based on their own reproduction. The results for T-Loss (Franceschi et al., 2019) are taken from the T-Loss paper directly (Franceschi et al., 2019).

For the anomaly detection tasks, the baseline results of SPOT, DSPOT (Siffer et al., 2017), and SR (Ren et al., 2019) are taken directly from the TS2Vec paper (Yue et al., 2022).

## A.4   TIME-EMBEDDING CHOICE

The choice of architecture for the time-embedding varies depending on the temporal features one wishes to extract. For users concerned with capturing periodicity and trend, a Time2Vec-like module is ideal (Kazemi et al., 2019). Focusing on these features has helped it become the top performer for anomaly detection and forecasting. If unsure about what features to extract, a fully-connect MLP may also be used, leading to perhaps less interpretable but very flexible and expressive embeddings. We found this to be the best time-embedding for classification tasks. We have also implemented a time-embedding model based on RBF features, which performs well in anomaly detection. It is not well understood when and why different time-embeddings perform better than others, beyond the simple case of Time2Vec (Kazemi et al., 2019). We leave this as an area for future research.

## A.5   CONTEXTUAL CONSISTENCY

TS2Vec introduces a new pair-sampling method for contrastive learning: *contextual consistency*. The idea is to sample two overlapping segments of the time-series and encourage the representations in the overlapping timesteps to be similar. Random transformations are applied to each sampled segment, leading to different *contexts* for the overlapping window. Ideally, the encoder should extract a representation of the window which is invariant to the surrounding context.

Given an input time-series $\mathbf{x} \in \mathbb{R}^{T \times F}$, random cropping is applied to obtain two overlapping segments of $\mathbf{x}$: $\mathbf{x}_{a_1:b_1}$ and $\mathbf{x}_{a_2:b_2}$ such that $0 \leqslant a_1 \leqslant a_2 \leqslant b_1 \leqslant b_2 \leqslant T$. $a_1, a_2, b_1, b_2$ are sampled uniformly in the allowed ranges at each training iteration. Timestamp masking is then randomly applied within each segment by the encoder, which outputs the respective representations $\mathbf{z}_{a_1:b_1} = \{z_{a_1}, ..., z_{b_1}\}$ and $\mathbf{z}_{a_2:b_2} = \{z_{a_2}, ..., z_{b_2}\}$. These representations in the overlapping interval $[a_2, b_1]$ correspond to the same timesteps, but were obtained using different contexts. Timestep-wise and instance-wise contrasting tasks are then applied to them Yue et al. (2022).

The contextual consistency method is very powerful as it helps learn invariances to different sub-sequence distributions (which might differ in the case of non-stationary data), without making any underlying assumptions about the time-series distribution. This contrasts other SSL methods for temporal data, which often feature inductive biases in the form of invariance assumptions to noise, cropping, flipping etc. (Zhang et al., 2023).

## A.6   HIERARCHICAL LOSS

The hierarchical loss framework applies and sums the model's loss function at different scales, starting from a per-timestep representation and applying `maxpool` operations to reduce the time-dimension between scales (Yue et al., 2022). This gives users control over the granularity of the representation used for downstream tasks, without sacrificing performance. It also makes the model more robust to missing data, as it makes use of long-range information in the surrounding representations to reconstruct missing timesteps (Yue et al., 2022). Its precise algorithm is given in Figure 5 below, taken from Yue et al. (2022):

---

**Algorithm 1:** Calculating the hierarchical contrastive loss

1: **procedure** HIERLOSS($r, r'$)
2:     $\mathcal{L}_{hier} \leftarrow \mathcal{L}_{dual}(r, r')$;
3:     $d \leftarrow 1$;
4:     **while** time_length($r$) $> 1$ **do**
5:         *// The maxpool1d operates along the time axis.*
6:         $r \leftarrow$ maxpool1d($r$, kernel_size $= 2$);
7:         $r' \leftarrow$ maxpool1d($r'$, kernel_size $= 2$);
8:         $\mathcal{L}_{hier} \leftarrow \mathcal{L}_{hier} + \mathcal{L}_{dual}(r, r')$ ;
9:         $d \leftarrow d + 1$ ;
10:     **end while**
11:     $\mathcal{L}_{hier} \leftarrow \mathcal{L}_{hier}/d$ ;
12:     **return** $\mathcal{L}_{hier}$
13: **end procedure**

Figure 5: Hierarchical loss algorithm (Yue et al., 2022)

## A.7 UEA CLASSIFICATION FULL RESULTS

Table 5 presents the full results for the classification task on the 30 UEA multivariate time series datasets Dau et al. (2019). The evaluated models are T-Rep, following the *TimeDim* procedure detailed in Appendix A.2.4, TS2Vec Yue et al. (2022), following both the *TimeDim* procedure and the original procedure introduced in Franceschi et al. (2019), T-Loss Franceschi et al. (2019), TNC Tonekaboni et al. (2021), TS-TCC Eldele et al. (2021) and DTW Müller (2007).

| Dataset | **T-Rep** (Ours) | TS2Vec | TS2Vec (*TimeDim*) | T-Loss | TNC | TS-TCC | DTW | Minirocket |
|---|---|---|---|---|---|---|---|---|
| ArticularyWordRecognition | 0.968 | 0.974 | 0.969 | 0.943 | 0.973 | 0.953 | 0.987 | 0.980 |
| AtrialFibrillation | 0.354 | 0.287 | 0.313 | 0.133 | 0.133 | 0.267 | 0.2 | 0.2 |
| BasicMotions | 1.0 | 1.0 | 1.0 | 1.0 | 1.0 | 1.0 | 1.0 | 1.0 |
| CharacterTrajectories | 0.989 | 0.988 | 0.988 | 0.993 | 0.967 | 0.985 | 0.989 | 0.988 |
| Cricket | 0.958 | 0.953 | 0.938 | 0.972 | 0.958 | 0.917 | 1.0 | 0.972 |
| DuckDuckGeese | 0.457 | 0.434 | 0.368 | 0.65 | 0.46 | 0.38 | 0.6 | 0.540 |
| ERing | 0.943 | 0.899 | 0.936 | 0.133 | 0.852 | 0.904 | 0.133 | 0.933 |
| EigenWorms | 0.884 | 0.916 | 0.88 | 0.84 | 0.84 | 0.779 | 0.618 | 0.954 |
| Epilepsy | 0.97 | 0.963 | 0.962 | 0.971 | 0.957 | 0.957 | 0.964 | 1.0 |
| EthanolConcentration | 0.333 | 0.303 | 0.298 | 0.205 | 0.297 | 0.285 | 0.322 | 0.357 |
| FaceDetection | 0.581 | 0.541 | 0.577 | 0.513 | 0.536 | 0.544 | 0.529 | 0.569 |
| FingerMovements | 0.495 | 0.495 | 0.493 | 0.58 | 0.47 | 0.46 | 0.53 | 0.420 |
| HandMovementDirection | 0.536 | 0.418 | 0.527 | 0.351 | 0.324 | 0.243 | 0.231 | 0.405 |
| Handwriting | 0.414 | 0.463 | 0.424 | 0.451 | 0.249 | 0.498 | 0.286 | 0.241 |
| Heartbeat | 0.725 | 0.734 | 0.724 | 0.741 | 0.746 | 0.750 | 0.717 | 0.722 |
| InsectWingbeat | 0.328 | 0.323 | 0.327 | 0.156 | 0.469 | 0.264 | 0.1 | 0.319 |
| JapaneseVowels | 0.962 | 0.97 | 0.961 | 0.989 | 0.978 | 0.93 | 0.949 | 0.921 |
| LSST | 0.526 | 0.558 | 0.546 | 0.509 | 0.595 | 0.474 | 0.551 | 0.668 |
| Libras | 0.829 | 0.859 | 0.833 | 0.883 | 0.817 | 0.822 | 0.87 | 0.944 |
| MotorImagery | 0.495 | 0.5 | 0.497 | 0.58 | 0.5 | 0.61 | 0.5 | 0.470 |
| NATOPS | 0.804 | 0.897 | 0.824 | 0.917 | 0.911 | 0.822 | 0.883 | 0.916 |
| PEMS-SF | 0.8 | 0.772 | 0.794 | 0.675 | 0.699 | 0.734 | 0.711 | 0.896 |
| PenDigits | 0.971 | 0.977 | 0.975 | 0.981 | 0.979 | 0.974 | 0.977 | 0.974 |
| PhonemeSpectra | 0.232 | 0.243 | 0.228 | 0.222 | 0.207 | 0.252 | 0.151 | 0.280 |
| RacketSports | 0.883 | 0.893 | 0.865 | 0.855 | 0.775 | 0.816 | 0.802 | 0.875 |
| SelfRegulationSCP1 | 0.819 | 0.79 | 0.819 | 0.843 | 0.799 | 0.823 | 0.775 | 0.884 |
| SelfRegulationSCP2 | 0.591 | 0.554 | 0.563 | 0.539 | 0.55 | 0.532 | 0.539 | 0.494 |
| SpokenArabicDigits | 0.994 | 0.992 | 0.993 | 0.905 | 0.934 | 0.97 | 0.963 | 0.989 |
| StandWalkJump | 0.441 | 0.407 | 0.3 | 0.332 | 0.4 | 0.332 | 0.2 | 0.333 |
| UWaveGestureLibrary | 0.885 | 0.875 | 0.89 | 0.875 | 0.759 | 0.753 | 0.903 | 0.913 |
| Avg. Accuracy | 0.706 | 0.699 | 0.693 | 0.657 | 0.671 | 0.667 | 0.650 | **0.707** |
| Ranks $1^{st}$ | 6 | 2 | 1 | 6 | 1 | 3 | 2 | **9** |
| Avg. Acc. Difference (%) | – | 2.1 | 3.4 | 33.6 | 12.3 | 10.0 | 43.6 | 4.8 |

Table 5: Classification accuracy of T-Rep and other self-supervised models for time series on 30 UEA datasets.

The metrics used for evaluation are the following:

- **Avg. Accuracy:** The measured test set accuracy, averaged over all datasets.

- **Ranks** $1^{st}$**:** This metric measures the number of datasets for which the given model is the best amongst all compared models. For example, T-Rep is the best performing model on 10 out of 30 datasets.

- **Avg. Acc. Difference:** This measures the relative difference between T-Rep and a given model $M$, averaged over all 30 datasets. It is calculated as:

$$\text{Avg. difference} = 100 \cdot \left( \sum_{i=1}^{30} \frac{Acc(\text{T-Rep})_i - Acc(M)_i}{Acc(M)_i} \right), \quad (11)$$

where $Acc(M)_i$ is the accuracy of model $M$ on dataset $i$.

- **Avg. Rank:** For each dataset, we compute the rank of a model compared to other models based on accuracy. The average rank over all 30 datasets is then reported.

## A.8 ETT FORECASTING FULL RESULTS

Table 6 presents the full results for the forecasting task on the 4 ETT datasets Zhou et al. (2021). We evaluate T-Rep against TS2Vec (Yue et al., 2022), a SOTA self-supervised model for time series, but also Informer (Zhou et al., 2021), the SOTA in time series forecasting, as well as TCN (Bai et al., 2018), a simpler model with the same backbone architecture as T-Rep, but trained with a supervised MSE loss.

| Dataset | H | **T-Rep** (Ours) MSE | **T-Rep** (Ours) MAE | TS2Vec MSE | TS2Vec MAE | Informer MSE | Informer MAE | TCN MSE | TCN MAE | Linear Baseline MSE | Linear Baseline MAE |
|---|---|---|---|---|---|---|---|---|---|---|---|
| ETTh$_1$ | 24 | **0.511** | **0.496** | 0.575 | 0.529 | 0.577 | 0.549 | 0.767 | 0.612 | 0.873 | 0.664 |
| | 48 | **0.546** | **0.524** | 0.608 | 0.553 | 0.685 | 0.625 | 0.713 | 0.617 | 0.912 | 0.689 |
| | 168 | **0.759** | **0.649** | 0.782 | 0.659 | 0.931 | 0.752 | 0.995 | 0.738 | 0.993 | 0.749 |
| | 336 | **0.936** | **0.742** | 0.956 | 0.753 | 1.128 | 0.873 | 1.175 | 0.800 | 1.085 | 0.804 |
| | 720 | **1.061** | **0.813** | 1.092 | 0.831 | 1.215 | 0.896 | 1.453 | 1.311 | 1.172 | 0.863 |
| ETTh$_2$ | 24 | 0.560 | 0.565 | **0.448** | 0.506 | 0.720 | 0.665 | 1.365 | 0.888 | 0.463 | **0.498** |
| | 48 | 0.847 | 0.711 | 0.685 | 0.642 | 1.457 | 1.001 | 1.395 | 0.960 | **0.614** | **0.588** |
| | 168 | 2.327 | 1.206 | 2.227 | 1.164 | 3.489 | 1.515 | 3.166 | 1.407 | **1.738** | **1.016** |
| | 336 | 2.665 | 1.324 | 2.803 | 1.360 | 2.723 | 1.340 | 3.256 | 1.481 | **2.198** | **1.173** |
| | 720 | 2.690 | 1.365 | 2.849 | 1.436 | 3.467 | 1.473 | 3.690 | 1.588 | **2.454** | **1.290** |
| ETTm$_1$ | 24 | 0.417 | 0.420 | 0.438 | 0.435 | **0.323** | **0.369** | 0.324 | 0.374 | 0.590 | 0.505 |
| | 48 | 0.526 | 0.484 | 0.582 | 0.553 | 0.494 | 0.505 | **0.477** | **0.450** | 0.813 | 0.637 |
| | 96 | **0.573** | **0.516** | 0.602 | 0.537 | 0.678 | 0.614 | 0.636 | 0.602 | 0.866 | 0.654 |
| | 288 | **0.648** | **0.577** | 0.709 | 0.610 | 1.056 | 0.786 | 1.270 | 1.351 | 0.929 | 0.697 |
| | 672 | **0.758** | **0.649** | 0.826 | 0.687 | 1.192 | 0.926 | 1.381 | 1.467 | 1.008 | 0.746 |
| ETTm$_2$ | 24 | 0.172 | 0.293 | 0.189 | 0.310 | **0.147** | **0.277** | 1.452 | 1.938 | 0.275 | 0.364 |
| | 48 | 0.263 | 0.377 | **0.256** | **0.369** | 0.267 | 0.389 | 2.181 | 0.839 | 0.363 | 0.434 |
| | 96 | 0.397 | 0.470 | 0.402 | 0.471 | **0.317** | **0.411** | 3.921 | 1.714 | 0.441 | 0.484 |
| | 288 | 0.897 | 0.733 | 0.879 | 0.724 | 1.147 | 0.834 | 3.649 | 3.245 | **0.754** | **0.664** |
| | 672 | 2.185 | 1.144 | 2.193 | 1.159 | 3.989 | 1.598 | 6.973 | 1.719 | **1.796** | **1.027** |
| Avg. Rank | | **1.90** | **1.85** | 2.40 | 2.45 | 3.3 | 3.55 | 4.35 | 4.1 | 3.05 | 3.05 |
| Ranks 1$^{st}$ | | **8** | **8** | 2 | 1 | 3 | 3 | 1 | 1 | 6 | 7 |
| Avg. MSE | | **0.986** | **0.702** | 1.004 | 0.712 | 1.300 | 0.820 | 2.012 | 1.205 | 1.017 | 0.727 |

Table 6: Multivariate time series forecasting results on the four ETT datasets. The 'Ranks $1^{st}$' metric measures the number of situations where the given model is the best amongst all compared models.

## A.9 Time Series Anomaly Detection Full Results

|               | $F_1$ | Prec. | Rec.  |
| ------------- | ----- | ----- | ----- |
| SPOT          | 0.338 | 0.269 | 0.454 |
| DSPOT         | 0.316 | 0.241 | 0.458 |
| SR            | 0.563 | 0.451 | 0.747 |
| TS2Vec        | 0.733 | 0.706 | 0.762 |
| **T-Rep** (Ours) | **0.757** | **0.796** | 0.723 |

Table 7: Time series anomaly detection results, on the Yahoo Webscope dataset. Anomalies include outliers as well as changepoints. TS2Vec results are reproduced using the official source code (Zhihan Yue, 2021), while all other baseline results are taken directly from Yue et al. (2022).

|               | $F_1$ | Acc.  |
| ------------- | ----- | ----- |
| Raw data      | 0.241 | 0.676 |
| TS2Vec        | 0.619 | 0.771 |
| **T-Rep** (Ours) | **0.665** | **0.790** |

Table 8: Time series segment-based anomaly detection results on the Sepsis dataset Reyna et al. (2020b). TS2Vec results are reproduced using the official source code (Zhihan Yue, 2021).

