# OpenReview forum: "T-Rep: Representation Learning for Time Series using Time-Embeddings"
_ICLR.cc/2024/Conference — ICLR 2024 poster_

### Official Review · Reviewer_9kuw · 2023-10-31

**Soundness:** 2 fair
**Presentation:** 3 good
**Contribution:** 3 good
**Rating:** 6
**Confidence:** 4

**Summary:**

This paper proposes a time embedding method in representation learning on unlabeled time series data. Following existing contrastive learning framework, it proposes two new regressive tasks to induce the learned time embeddings to play a good role in the final representation. Experiments on three different task categories  are performed to support the effectiveness of the proposed designs. An analysis on the robustness to missing data is also provided.

**Strengths:**

1. The presentation of the paper is smooth and easy to follow. The limitation of the existing methods and the novel part in the proposed method is quite clear.
2. Comprehensive experiment results are provided to show the versatility of the learned representations.

**Weaknesses:**

1. The biggest flaw of the paper is the lack of ablation study, in my opinion. While sufficient quantitative results are provided to showcase the advantage of the proposed T-Rep, there is no clear evidence that support the effectiveness of the time embedding module, the very core highlighted in the paper. While the evidence in the analysis of robustness to missing data claims that the use of time embedding is leveraged to improve the contextual awareness and to fill the gap of the missing interval, but it is not supported by any rigorous analyses and remains a conjecture.
Extensive ablation studies should be provided to compare the T-Rep with the baseline of TS2Vec and versions with either of the regressive pretext task removed.
2. Above the flaw in empirical study, the idea of using time embedding to improve contrastive learning is not well-motivated. It is not clear why time embedding would be the ideal way to improve the latent temporal structure. See details in questions.

**Questions:**

1. While the authors claim that time embedding $\tau$ are induced to learn such time-related features as trend, periodicity etc. , they are function of the time indices and are unaware of the time series values. How could the aforementioned features be captured from indices only?
2. I understand that $\tau$ is normalized so that their difference can be measured by JSD. But there are alternative ways to measure the divergence without confining the norm of time embeddings.
3. The task in Sec 4.2.1 aims to regress difference in the representation to the time distance. As stated in point 1, again, various pattern difference might be contained in the representations from arbitrary pairs of time series, therefore it's very likely that they can not regress to the consistent time distance. Same concern applies to the other task
4. The claim that the two proposed task complement each other needs to be elaborated.

---

> ### Author Response · Authors · 2023-11-16
> **Response to Rewiewer 9kuw - Part 1**
>
> We sincerely appreciate the positive feedback regarding the presentation of our paper. It is gratifying to hear that the content is smooth and easy to follow. We are also pleased that the limitations of existing methods and the novelty of our proposed approach are clearly communicated. We totally agree that the biggest flaw of the paper is the lack of ablation study, as explained in the global answer we are working on it and we will update the submission with these results in a few days.
>
> **Q1. While the authors claim that time embedding tau are induced to learn such time-related features as trend, periodicity etc. , they are function of the time indices and are unaware of the time series values. How could the aforementioned features be captured from indices only?**
>
> **A1.** T-Rep is a parameterised function we are learning, containing a sub-module for learning time-embeddings from time indices, an a module for feature extraction from the time series, with all modules interconnected. Because the loss function used to train T-Rep is a joint function of the produced representations and time-embeddings, the loss backpropagates through the feature extractor as well as the time-embedding module. This makes the time-embedding ‘aware’ of the time series values, through the loss function’s gradient (the loss is a function of the time series and the time indices).
>
> The loss function is given by the pretext tasks, which have been designed to encourage the time-embeddings to contain meaningful information about the time-series (particularly the ‘Time-embedding conditioned forecasting’ task, see section 4.2.2 for more details). This consequently pushes the time-embedding to contain time-related features.
>
> Now the nature of the features (e.g. trend, periodicity etc.) is, we believe, not only influenced by the pretext tasks but also by the choice of time-embedding architecture. For example, for forecasting and anomaly detection, we use Time2Vec (https://arxiv.org/abs/1907.05321). This architecture is quite interpretable: the first component of the time-embedding vector is a linear function, which Time2Vec authors claim learns the trend of the time-series. The other components are sinusoidal features with learned period and phase-shift, aiming to capture periodicity in the data at different scales.
>
> We hope this addresses your question. If you want more details, we encourage you to look at Time2Vec (https://arxiv.org/abs/1907.05321), which was the first paper to introduce this idea of learning a time-embedding module or function alongside the feature extractor/predictor.
>
> **Q2. I understand that tau is normalized so that their difference can be measured by JSD. But there are alternative ways to measure the divergence without confining the norm of time embeddings.**
>
> **A2.** Thank you for this very relevant question, which we thoroughly addressed during the model design process. Indeed, there are alternative ways to measure divergence without restricting the norm of time embeddings, but we haven't found any that provide as much satisfaction as the Jensen-Shannon Divergence (JSD). We explored the use of Kullback-Leibler Divergence, but it has some significant drawbacks: it is not symmetric, sensitive to zero values, and not defined on disjoint distributions. We also attempted to employ Wasserstein distance, but it proved computationally expensive, especially in high-dimensional spaces, leading us to quickly dismiss this option. The JSD appeared to be the best compromise, but we are open to suggestions if you believe that alternative approaches would be more relevant.

---

> > ### Author Response · Authors · 2023-11-16
> > **Response to Reviewer 9kuw - Part 2**
> >
> > **Q3. The task in Sec 4.2.1 aims to regress difference in the representation to the time distance. As stated in point 1, again, various pattern difference might be contained in the representations from arbitrary pairs of time series, therefore it's very likely that they can not regress to the consistent time distance. Same concern applies to the other task**
> >
> > **A3.** This observation is correct, it’s highly unlikely that we regress perfectly according to the time-embedding divergence (or any other regression target for that matter), given we have multiple pretext tasks. However, we argue this is a good thing. Our goal is to learn **general** representations of time-series, containing a variety of features: spatial, temporal, features that help discriminate instances, others more focused on intra-instance nuances etc. By having multiple pretext tasks, there is a **regularisation** effect on the representations. The loss of each pretext task will be slightly higher than in the single-task case, due to the competing features needed for each task, but their lower individual contribution to the total loss encourages a **general** optimum to be found, minimising the overall loss achieved. Empirically, this results representations which contain features required for the different pretext tasks. These representations are thus more general and robust. Please see https://arxiv.org/pdf/2306.10125.pdf and https://arxiv.org/abs/2303.01034 for more details.
> >
> > As an example, consider the task of Sec 4.2.1. If we were to regress the temporal divergence perfectly, this would infer that the representation vector contains almost only temporal features, which is undesirable. However, if we manage to regress the temporal divergence approximatively, and are simply proportional to the time distance, this indicates that we have incorporated the notion of temporality in our representations, along with other features. We can be confident these other features are not random noise because of how we designed the other pretext tasks which are jointly optimised.
> >
> > **Q4. The claim that the two proposed task complement each other needs to be elaborated.**
> >
> > **A4.** We completely agree with you, and as you rightly pointed out, the biggest flaw of the paper is the lack of an ablation study. We are currently running experiments to demonstrate that the combination of both tasks yields good results, whereas a single task produces inferior results compared to no task at all.
> >
> > Having started the ablation study, we can already provide you with results of the ablation study on the forecasting task. We observe that the two pretext tasks complement each other well: if only one of them is used, the performance is lower than using none, and significantly lower than using both. Additionally, we can see that the time-embedding (TE) module effectively enhances performance, and it is the combination of these elements (TE + 2 proposed pretext tasks) that allows us to outperform the state-of-the-art TS2Vec.
> >
> > |                                   | Forecasting |
> > |-----------------------------------|-------------|
> > |                                   | Avg. MSE    |
> > |-----------------------------------|-------------|
> > | **T-Rep**                          | **0.986**   |
> > |-----------------------------------|-------------|
> > | *Pretext tasks*                    |             |
> > | w/o TE-conditioned forecasting    | 1.022 (+3.7%) |
> > | w/o TE divergence prediction       | 1.003 (+1.7%) |
> > | w/o New pretext tasks               | 0.999 (+1.3%) |
> > |-----------------------------------|-------------|
> > | *Architecture*                     |             |
> > | w/o TE module (=TS2Vec)            | 1.004 (+1.8%) |
> >
> > *Ablation results on ETT forecasting datasets and the {} anomaly detection dataset. The percentage changes are calculated as the relative difference between the modified model's performance and T-Rep's performance.*
> >
> > *If our responses and revisions meet your expectations, we would greatly appreciate it if you could consider revising your evaluation accordingly. Thank you for your time and thorough review.*

---

> > > ### Author Response · Authors · 2023-11-22
> > > **Response to Rewiewer 9kuw - Experiments**
> > >
> > > We have now conducted the ablation study you requested, which you can find in section 5.5 of the paper in blue. The results confirm that the proposed pretext tasks and the addition of a time-embedding module to the encoder contribute to T-Rep’s performance: removing any of these decreases the scores in both tasks. These results also illustrate the interdependency of both tasks, as in forecasting, only leaving one of the tasks obtains worse results than removing both pretext tasks. It also justifies our preferred choice of time-embedding, since Time2Vec (Kazemi et al., 2019) outperforms the other 2 architectures in both tasks.
> > >
> > > Thank you for proposing these improvements.
> > >
> > > *If our adjustments meet your expectations, we would greatly appreciate it if you could consider revising your evaluation accordingly. Thank you for your time and thorough review.*

---

> > > > ### Comment · Reviewer_9kuw · 2023-11-22
> > > > **Thanks for the detailed response.**
> > > >
> > > > I appreciate author's detailed response to my questions. Overall my concern is mostly about the ablation study, and as it has been provided, I am happy to adjust my rating.
> > > >
> > > > Regarding the answers to generalizability of time embedding (Q1 and Q3), I agree that periodicity of a specific task/dataset can be fitted with Time2Vec as it learns to project a periodical time series onto a set of Fourier bases. However, trend is not easy, if it's not impossible at all, to be fitted given the great variety by a single linear term. In fact, the original Time2Vec paper (arxiv: 1907.05321) did not articulate the ability of learning "trends". They only claim the linear term, i.e. the first dimension of the embedding, "can model nonperiodic components and helps with extrapolation". Therefore I believe "capturing trends" in this paper is somewhat over-claimed.

---

> > > > > ### Author Response · Authors · 2023-11-23
> > > > >
> > > > > Dear reviewer,
> > > > > Thank you for reviewing our responses and the revised paper. We appreciate that you found the new experiments satisfactory and adjusted the grade accordingly.
> > > > > You are right, Time2Vec is only able to extract the linear component of trend and not the entire trend, an important nuance. However, we believe that since the time-embedding is only one component of our much larger encoder (including a 10 block 1D convolution resnet), the combination of these different elements allows our model to capture more than just the linear component of trend (even if not the entire trend). This is in part supported by (Bai. et al, 2018), which the ability of 1d convolutions to extract features from time series.
> > > > > Nonetheless, we have not empirically verified this and it is highly likely that T-Rep cannot capture all the trend components of the dataset, so we will adapt the paper to be less confident in claims about capturing trend. We hope this addresses your concerns and thank you again for taking the time to review our paper.

---

### Official Review · Reviewer_5x3c · 2023-11-02

**Soundness:** 3 good
**Presentation:** 4 excellent
**Contribution:** 3 good
**Rating:** 8
**Confidence:** 3

**Summary:**

The paper introduces a sef-supervised representation learning method with a temporal, timestep granularity. The model is a linear projection layer and time embedding module that is fed into a dilated convolutional encoder. This representation is then used by four pretext tasks, two of which (time-embedding divergence prediction and time-embedding conditioned forecasting) are introduced in this paper. A linear combination of the pretext task losses is then used to compute a hierarchical loss. The paper evaluates the approach on different downstream tasks, including anomaly detection, forecasting, and classification. Additionally, the model authors show that the model is robust to missing data and qualitatively visualize the time evolution of the representation.

**Strengths:**

### Originality
Introducing the temporal structure into time series representations are an important problem. This paper attempts to capture the time evolution with two additional pretext tasks and a time embedding module. These components are well reasoned and appear to result into a representation that can be successfully leveraged in tasks that require timestep granularity (forecasting, anomaly detection) and tasks that might not necessarily need it (classification).

### Quality
The discussion of the related work covers related articles well and summarizes the gap in learning temporal resolution in representation learning for time series. The experimental section covers standard benchmarks for anomaly detection, forecasting, and classification. I think the experimental evaluation has some weaknesses, which I will elaborate in the Weaknesses section.

### Clarity
The paper is well written. The introduction and related work clearly introduces the challenges and previous work in this space and points out the specific challenge that the paper aims to address (learning timestep granular representations that work for several downstream tasks). The experimental section is also well written and easy to follow.

### Significance
Time series representation learning is an important problem. Previous work focused on classification or other tasks that don't require timestep granularity of the embeddings, while some applications (like forecasting) would require this. This paper represents a significant step into this direction of universal time series embeddings that are useful for several different applications.

**Weaknesses:**

The paper has two weaknesses in the experimental evaluation: Missing ablation experiments and choice of baselines.

### Ablation study

The method presented in this work shares several components with TS2Vec (linear projection layer, dilated convolution encoder, instance-wise/temporal contrasting, and the hierarchical loss). It would be beneficial for the readers to point out the common components of T-Rep and TS2Vec. It would also be interesting for the reader to understand, which of the additional components introduced in this work actually improved the performance, but an ablation study to investigate this is missing. My understanding is that the additional pretext tasks require the time embedding (TE) module, so factoring that out for ablation is probably difficult. However, testing the impact of the pretext tasks setting the weights of the linear combination loss for the TE-conditioned forecasting and/or TE-divergence prediction should be straightforward. I would kindly ask the authors to add these ablation results. I would consider raising my score if the ablation study is added.


### Choice of baselines

For the forecasting experiments, the authors use the benchmark introduced by Zhou et al., AAAI 2021 is a common benchmark in forecasting. However, several iterations on transformer architectures have been published since this work, some of which are even outperformed by linear baselines (Zeng et al., AAAI 2022). Given that the forecasting task introduced here is a linear regression layer using the time series embeddings, it would be good to understand the gain in performance relative to the linear methods (N-Linear and D-Linear) that are introduced in Zeng et al., AAAI 2022. I would kindly ask the authors to consider these baselines.

For classification, Minirocket (Dempster at al., KDD 2021) is a simple and fast baseline for practical applications and it would be interesting to understand the gain in accuracy from T-Rep over Minirocket.

**Questions:**

Appendix A.2 mentions that each models are run 10 times but also notes "Also, we do not use any random seed, (...)". I'm unsure what this means. Are all experiments started from the same seed or are ten (fixed) seeds used here? I would kindly ask the authors to clarify that.

---

> ### Author Response · Authors · 2023-11-16
> **Response to Reviewer 5x3c**
>
> We appreciate your very detailed review and the attention given to our submission. We are pleased that you found our presentation excellent and the paper easy to read. Regarding the ablation study, as mentioned in the overall response, we are in the process of finalizing it to add these results to the article because we completely agree that it is a crucial experiment to include in our submission.
> In the same manner, we are in the process of adding the baselines you suggest.
>
> Having started the ablation study, we can already provide you with results of the ablation study on the forecasting task. We observe that the two pretext tasks complement each other well: if only one of them is used, the performance is lower than using none, and significantly lower than using both. Additionally, we can see that the time-embedding (TE) module effectively enhances performance, and it is the combination of these elements (TE + 2 proposed pretext tasks) that allows us to outperform the state-of-the-art TS2Vec.
>
> |                                   | Forecasting |
> |-----------------------------------|-------------|
> |                                   | Avg. MSE    |
> |-----------------------------------|-------------|
> | **T-Rep**                          | **0.986**   |
> |-----------------------------------|-------------|
> | *Pretext tasks*                    |             |
> | w/o TE-conditioned forecasting    | 1.022 (+3.7%) |
> | w/o TE divergence prediction       | 1.003 (+1.7%) |
> | w/o New pretext tasks               | 0.999 (+1.3%) |
> |-----------------------------------|-------------|
> | *Architecture*                     |             |
> | w/o TE module (=TS2Vec)            | 1.004 (+1.8%) |
>
> *Ablation results on ETT forecasting dataset. The percentage changes are calculated as the relative difference between the modified model's performance and T-Rep's performance.*
>
>
>
> **Q1. Appendix A.2 mentions that each models are run 10 times but also notes "Also, we do not use any random seed, (...)". I'm unsure what this means. Are all experiments started from the same seed or are ten (fixed) seeds used here? I would kindly ask the authors to clarify that.**
>
> **A1.** Thank you for raising this point, which seems to lack clarity in our article and was also highlighted by reviewer rWLt. Instead of fixing a seed, which has the disadvantage of not accounting for the variance that may exist between two seeds, we chose to run each experiment between 10 and 20 times to address variance and obtain reliable results. The reported results represent the average of these runs. This approach maximizes the reliability and robustness of our results, avoiding the selection of a specific seed that might produce the best results. During our 10 or 20 runs, the seed is set to "None." When the seed is set to None, PyTorch utilizes the system time or another unpredictable source to initialize the random number generator. Consequently, each time you run the program, you are likely to get different random numbers.
>
> *If our responses and revisions meet your expectations, we would greatly appreciate it if you could consider revising your evaluation accordingly. Thank you for your time and thorough review.*

---

> > ### Author Response · Authors · 2023-11-22
> > **Response to Reviewer 5x3c - Experiments**
> >
> > We have now conducted the ablation study you requested, which you can find in section 5.5 of the paper in blue.
> > The results confirm that the proposed pretext tasks and the addition of a time-embedding module to the encoder contribute to T-Rep’s performance: removing any of these decreases the scores in both tasks. These results also illustrate the interdependency of both tasks, as in forecasting, only leaving one of the tasks obtains worse results than removing both pretext tasks. It also justifies our preferred choice of time-embedding, since Time2Vec (Kazemi et al., 2019) outperforms the other 2 architectures in both tasks.
> >
> > We also added the Minirocket baseline for the classification taskk, which you can find in section 5.2 of the paper in blue.
> > Results show that T-Rep beats TS2Vec by 2.1% and Minirocket by 4.8% on average, a significant margin. In terms of average accuracy, TRep’s accuracy outperforms all competitors except Minirocket, which has a 0.001 lead. T-Rep has a positive ’Avg. difference’ to Minirocket despite having a lower ’Avg. Acc.’ because Minirocket often performs slightly better than T-Rep, but when T-Rep is more accurate, it is so by a larger margin.
> > It is important to note that Minirocket was developed specifically for time series classification (and
> > is the SOTA as of this date), while T-Rep is a general representation learning model, highlighting its
> > strengths across applications. The latent space dimensionality is set to 320 for all baselines, except
> > Minirocket which uses 9996 dimensions (as per the official code). T-Rep uses only 128 dimensions,
> > but leverages the temporal dimension of its representations.
> >
> > Finally, we added a linear baseline for the forecasting task, which you can find in section 5.3 of the paper in blue.
> > Results confirm that T-Rep achieves the best average scores with a MSE of 0.986 and a MAE of 0.702 while the linear baseline has an MSE of 1.017 and a MAE of 0.727.
> >
> > Thank you for proposing these improvements.
> >
> > *If our adjustments meet your expectations, we would greatly appreciate it if you could consider revising your evaluation accordingly. Thank you for your time and thorough review.*

---

> > > ### Comment · Reviewer_5x3c · 2023-11-22
> > > **Response to Rebuttal**
> > >
> > > I would like to thank the authors for adding additional experiments and analysis for their paper. My points have been addressed and I will increase my score.
> > >
> > > However, I think with the additional baselines for forecasting and classification, it appears that the proposed model only performs slightly better (forecasting) and sometimes worse or better in classification (depending on the metric). In essence, the performance is close to baselines. The paper still has its merits in being a representation method that appears to work across tasks when compared to task-specific models.
> > >
> > > Given that the standard error/confidence interval is not accounted for in this paper to test whether the performance differences are statistically significant, I would suggest that the authors make sure the language chosen does not suggest that. For example: "Table 2 shows that T-Rep beats TS2Vec by 2.1% and Minirocket by `4.8% on average, a significant margin." suggests that some sort of significant testing is performed (which is not as far as I can tell). I think the paper still has its merits and the method is on-par with task-specific baselines, which I think is sufficient for publication.

---

> > > > ### Author Response · Authors · 2023-11-23
> > > >
> > > > Dear reviewer,
> > > >
> > > > Thank you very much for taking the time to read through our answers and updated paper.
> > > > We are very grateful you were satisfied with the new experiments and raised your grade accordingly. You make a good point, the statistical evidence of our results is not thoroughly investigated, although we did our best to obtain reliable results by running each experiment ten times, to eliminate as much stochasticity as possible.
> > > > We have updated the paper to be less assertive when presenting the results of the various experiments, mentioning these are only slight improvements. We hope this addresses your concerns and thank you again for taking the time to review our paper.

---

### Official Review · Reviewer_znEu · 2023-11-03

**Soundness:** 3 good
**Presentation:** 3 good
**Contribution:** 2 fair
**Rating:** 5
**Confidence:** 5

**Summary:**

The paper proposes T-Rep, a self-supervised method for learning time series representations at the timestep level. The key innovation of T-Rep is the use of learnable time embeddings in pretext tasks to learn detailed temporal dependencies and robustness in time series representations. Experiments demonstrate improved performance.

**Strengths:**

The paper proposes T-Rep, a self-supervised method for learning representations of time series at the timestep level.

T-Rep learns vector embeddings of time called "time-embeddings" alongside its feature extractor encoder. The time-embeddings help capture temporal features like trend, periodicity, distribution shifts.

The time-embeddings are incorporated into pretext tasks to learn fine-grained temporal dependencies and make the model robust to missing data. Two new pretext tasks are proposed: time-embedding divergence prediction and time-embedding-conditioned forecasting.

T-Rep is evaluated on downstream tasks of classification, forecasting, and anomaly detection. It outperforms previous self-supervised methods like TS2Vec, showing the benefit of time-embeddings.

T-Rep is more robust to missing data than methods like TS2Vec. Visualizations show T-Rep can produce smooth representations even with missing timesteps.

**Weaknesses:**

The choice of time-embedding architecture is not well motivated or analyzed. Different architectures are used for different tasks, but it is unclear why they perform best in each case. More ablation studies on the time-embedding design could strengthen this key component.

The pretext tasks using time-embeddings seem somewhat ad-hoc. While they demonstrate the utility of time-embeddings, developing more principled pretext tasks derived from intrinsic properties of time series could be beneficial.

The comparison to previous methods like TS2Vec is not entirely fair, as T-Rep uses a larger encoder architecture. Comparisons with a TS2Vec model of comparable complexity could better isolate the benefits of the time-embeddings. Besides, the composition on TimeNet and PatchTST on SOTA ETT dataset is necessary.

The treatment of missing data is a major claimed contribution, but the missing data experiments are limited. More systematic tests on real-world messy data with different missing data types could better showcase these abilities.

The interpretability of representations is claimed but only briefly demonstrated with some visualizations. More analysis connecting latent dimensions to meaningful time series properties could better support the interpretability claims.

The classification task does not standalone evaluate the quality of the learned representations. Adding unsupervised evaluations like clustering could help assess representation quality.

**Questions:**

See the weekness.

**Details Of Ethics Concerns:**

-

---

> ### Author Response · Authors · 2023-11-16
> **Response to Reviewer znEu - Part 1**
>
> Thank you very much for all your highly relevant suggestions, motivating experiments that significantly enhance the quality of the article. We appreciate your considerations regarding interpretability and the quality of representations, which align with those of reviewers 99iS and rWLt.
> Currently, we are conducting experiments to demonstrate the meaningful time series properties of latent dimensions, and we will send you the results in the next few days.
> Similarly, we are incorporating comparisons with the baselines you suggest, and we are working on highlighting the robustness of our model to missing data, with a new real-world anomaly detection dataset.
>
> **Q1. The choice of time-embedding architecture is not well motivated or analyzed. Different architectures are used for different tasks, but it is unclear why they perform best in each case. More ablation studies on the time-embedding design could strengthen this key component.**
>
>
> **A1.** Thank you for pointing out this issue. The field of time-embeddings is very recent and little literature exists. The most famous time-embedding architecture is Time2Vec (https://arxiv.org/abs/1907.05321), which we use in forecasting and anomaly detection tasks. This architecture is actually relatively interpretable: the first component of the time-embedding vector is a linear function, which Time2Vec authors claim learns the trend of the time-series. The other components are sinusoidal features with learned period and phase-shift, aiming to capture periodicity in the data at different scales. This focus on learning a dataset’s inherent trend and periodicity explains why it helps the T-Rep outperform its counterparts in the forecasting and anomaly detection tasks.
> Generally speaking, constrained time-embedding architectures (i.e. sinusoidal form, linear, RBF features) will tend to be more interpretable than their very expressive but more ‘black-box’ MLP-based counterparts.
> We believe MLP-based time-embeddings outperform sinusoidal ones in classification (and other instance-wide tasks) because they can capture more ‘global’ and coarse features of a time-series instance’s temporal structure, allowing T-Rep differentiate different instances of a dataset more easily in downstream tasks. This is a result of the expressivity of MLPs and the lack of constraint on the features they can extract, compared to sinusoidal or RBF-based time-embedding architectures. It is very hard to give more rigorous explanations, as the learned features are not human-interpretable.
> Overall, we agree that there is limited understanding of why time-embeddings behave differently depending on the task. This is because it is a new niche of machine learning which has not been studied in great depth, and few people have experimented with thus far (we haven’t come across any papers using them in classification).
>
> **Q2. The pretext tasks using time-embeddings seem somewhat ad-hoc. While they demonstrate the utility of time-embeddings, developing more principled pretext tasks derived from intrinsic properties of time series could be beneficial.**
>
> **A2.** Thank you for your highly relevant question, which happens to be one of the main avenues of our future work. We completely agree that these pretext tasks are ad-hoc, and our intention was to experimentally verify their relevance before delving into the theoretical foundation of these pretext tasks using time embeddings. We are particularly interested in this aspect, which will be the focus of our upcoming work.
>
> **Q3. The comparison to previous methods like TS2Vec is not entirely fair, as T-Rep uses a larger encoder architecture. Comparisons with a TS2Vec model of comparable complexity could better isolate the benefits of the time-embeddings. Besides, the composition on TimeNet and PatchTST on SOTA ETT dataset is necessary.**
>
> **A3.** That is absolutely true, thank you for noting this point. We use a layer of 128 instead of 64 to facilitate the concatenation of the time embedding to the time series (an issue TS2Vec does not have to deal with). We considered that these dimensions, while larger, are still "small" (we are not in the range of 2048, for instance), making it acceptable.
>
> We have begun implementing the pipeline to compare PatchTST on the state-of-the-art ETT forecasting dataset. This will be included in the final version of the article, and we hope to present the results to you by the end of the discussion period. Based on our understanding, TimeNet cannot be included in this comparison as it has been developed for classification, and not for tasks requiring timestep granularity, such as forecasting. Do you think this comparison is really necessary despite this aspect?

---

> > ### Author Response · Authors · 2023-11-16
> > **Responsed to Reviewer znEu - Part 2**
> >
> > **Q4. The treatment of missing data is a major claimed contribution, but the missing data experiments are limited. More systematic tests on real-world messy data with different missing data types could better showcase these abilities.**
> >
> > **A4.** You are absolutely correct, and this point is recurrent among various reviewers. We agree on the importance of adding an experiment with real-world messy data. To address your comment and the one from the reviewers 99iS and rWLt, we are currently replicating our experiments on the 'Sepsis' dataset from The PhysioNet/Computing in Cardiology Challenge 2019, which contains multivariate data sourced from 40 000 Intensive Care Unit patients in three separate hospital systems. We chose this dataset because it contains real-life data, includes missing data, and anomalies appear by segment (our other anomaly detection dataset, Yahoo, features point-based anomalies). Link of the dataset can be found here: https://journals.lww.com/ccmjournal/fulltext/2020/02000/early_prediction_of_sepsis_from_clinical_data__the.10.aspx.
> >
> > **Q5. The interpretability of representations is claimed but only briefly demonstrated with some visualizations. More analysis connecting latent dimensions to meaningful time series properties could better support the interpretability claims and 6. The classification task does not standalone evaluate the quality of the learned representations. Adding unsupervised evaluations like clustering could help assess representation quality.**
> >
> > **A5.** Thank you for this question and this additional experimentation proposition. There is not a direct mapping between the latent dimensions and the input features in a strict sense, even though we observe long-term trends on certain channels and short-term trends on others. The model does not possess this type of interpretability, as the learned features are not human-interpretable. Also, the training objectives used do not explicitly encourage features to be present in specific dimensions, we leave the liberty of allocating features to specific dimensions to the model. However, interpretability can be achieved within the clusters obtained through unsupervised evaluation to assess the quality of representations, as suggested in your **question 6**. We have set up everything for this experiment, including the other aspects requested by you and other reviewers, and hope to have the results as soon as possible. They will be included in the final version of the article regardless.
> >
> > *If our responses and revisions meet your expectations, we would greatly appreciate it if you could consider revising your evaluation accordingly. Thank you for your time and thorough review.*

---

> > > ### Author Response · Authors · 2023-11-22
> > > **Responsed to Reviewer znEu - Experiments**
> > >
> > > We have now conducted experiments on the Sepsis dataset, containing missing data, which you can find in section 5.1 of the paper in blue. The results confirm that T-Rep performs better than TS2Vec and a linear baseline on a real-world dataset.
> > >
> > > We have also conducted visualizations experiments to help assess representation quality, available in section A.1 of the paper in blue. This experiment truly highlights the versatile nature of T-Rep, improving the quality of representations at both the dataset scale (inter-instance discrimination) and individual time series (timestep granularity).
> > >
> > > The only result that we are still missing is that of the comparison with PatchTST on ETT dataset, and we apologize for this as we have not yet had the time to conclude this experiment.
> > >
> > > *If our adjustments meet your expectations, we would greatly appreciate it if you could consider revising your evaluation accordingly. Thank you for your time and thorough review.*

---

> > > > ### Comment · Reviewer_znEu · 2023-11-23
> > > > **Thanks for the response**
> > > >
> > > > Dear Authors,
> > > >
> > > > Thanks for your efforts on replying my comments. While most of the concerns are addressed, I cannot see any of the updates of my Q3. Note that the comparison with state-of-the-art works is important which will help to position this paper. I would like to hear all the potential helpful updates or discussions on this issue.
> > > >
> > > > Therefore, I tend to stick my score at current stage.
> > > >
> > > > Best

---

> > > > > ### Author Response · Authors · 2023-11-23
> > > > > **TimeNet and PatchTST**
> > > > >
> > > > > Dear reviewer, thank you for taking the time to go through our answers and updated paper. We are sorry the modifications do not meet your expectations.
> > > > >
> > > > > The reason we have not included TimeNet in our forecasting baselines is that the model was designed for classification and not forecasting ([[1706.08838] TimeNet: Pre-trained deep recurrent neural network for time series classification (arxiv.org)](https://arxiv.org/abs/1706.08838)), so it did not make sense to include it on the ETT dataset. Further, the model is dated from 2017, so we deem it unlikely to still be state of the art given the steady pace of progress in deep learning.
> > > > >
> > > > > As for PatchTST, we agree it is a very interesting baseline, and we have thus started working on the implementation to include the model in our baselines, but it is not a straightforward task. Firstly, the proposed Transformer architecture is very large and takes significant time to train on our hardware. Secondly, the pre-processing applied to the ETT datasets in their experiments differs from ours. Finally, the most important reason is that the authors of PatchTST use a different forecasting protocol by fine-tuning their transformer for the forecasting task after the self-supervised pre-training, whereas we simply add a linear regression model to our self-supervised representations. This means we need to make a lot of changes to the existing implementation in order to obtain results for PatchTST, and need to account for the training time. We will update the paper with the results as soon as we have them, but it is unlikely to be done by the deadline, which is in a few hours. However, this will be present in the camera-ready version of the article.
> > > > >
> > > > > We are very sorry we did not have time to obtaining results for PatchTST, which has been delayed by the addition of 3 other experiments to the paper: an extensive ablation study on forecasting and anomaly detection, a comparative clustering experiment, and a new real-world messy dataset for anomaly detection. Additionally, recognizing the importance of comparing ourselves to SOTA models, we have included 3 new baselines: a baseline training on raw data for the forecasting and anomaly detection tasks, and a new SOTA classification baseline (Minirocket, [[2012.08791] MINIROCKET: A Very Fast (Almost) Deterministic Transform for Time Series Classification (arxiv.org)](https://arxiv.org/abs/2012.08791)). The work and training times required for these various tasks have taken up a large part of the rebuttal period, delaying the experiments on PatchTST.
> > > > >
> > > > > We hope this answer provides some clarity on why the PatchTST and TimeNet results are not provided in the latest update of the paper, and thank you again for thoroughly reviewing our paper.

---

### Official Review · Reviewer_rWLt · 2023-11-04

**Soundness:** 3 good
**Presentation:** 4 excellent
**Contribution:** 3 good
**Rating:** 6
**Confidence:** 4

**Summary:**

The paper proposes a time series representation learning methodology based on self-supervision. The paper identifies two key issues with current time-series embedding using contrastive learning which -- a) aims to embed entire trajectories based on a binary notion of similarity vs. dissimilarity, ignoring temporal structure, and b) incompatibility with systems which switch between states, where notions of similarity vs. dissimilarity are based on states. They propose two pretext tasks to address these issues.


Overall, I think the paper is very well written and motivated, except a few things that can really improve the quality. I hope authors can address these in the discussion.

**Strengths:**

1. The paper is very well motivated, I congratulate the authors on explaining the issues with contrastive learning in the context of time-series.
2. The paper is overall well-written and easy to follow.
3. Experiments are set-up well except that monte-carlo simulations are missing.

**Weaknesses:**

1. The results are not repeated across random seeds, which significantly impacts the confidence in the method.
2. Visualizations: since the paper centers around time series representation learning, it would have been extremely valuable to see t-SNE plots of the learned embeddings to see how T-Rep performs over SOTA (TS2VEC) and others.
3. There are a number of moving parts, and some level of ablations are expected, but missing. For instance, impact of overlap in contexual consistency etc.

**Questions:**

1. How may dataset are used for anomaly detection (sec 5.1)?
2. How does the issues of similarity/dissimilarity play out in multivariate settings? What if the switching behaviour is only present in a few of the channels?
3. Why does T-Rep perform better in some cases and not it others? For instance, in Table 5 in ETTh_2 T-Rep performs better at larger horizons, while being significantly worse at lower ones. This is counter-intuitive, since in section 4.2.2. the paper mentions that the pretext forecasting task specifically aims to predict shorter horizons.
4. It seems that the methods requires supervision in pretest task 4.2.1 and not in 4.2.2 for the pretext task, can the authors comment on potential use of T-Rep in completely unsupervised pretraining?
5. How much overlap do we need for contextual consistency? How is this parameter decided?
6. In section 3, the terms such as contextual consistency and heirarchical loss are only defined intuitively. These need to be defined in terms of their mathematical formulation(s). Currently, these are difficult to understand. Furthermore, it is also unclear how this plays out mathematically in linear projection layer.
7. What is the mathematical action of TCN layer, and what is its role?


Minor:
1. Fig. 1 uses yellow against a whote background is difficult to read.

---

> ### Author Response · Authors · 2023-11-16
> **Response to Reviewer rWLt - Part 1**
>
> Thank you for taking the time to review our article, and we sincerely appreciate your thoughtful comments. Your positive remarks on the motivation and clarity of our paper are encouraging, and we are pleased that you found the overall presentation to be well-written and easy to follow. We are currently working on the final experiments for visualizations and the ablation study, as you recommended and as explained in the common answer. In the meantime, here are the answers to your various questions:
>
> **Q1. How many datasets are used for anomaly detection (sec 5.1)?**
> **A1.** The Yahoo Dataset, consisting of 367 synthetic and real univariate time series featuring outlier and change point anomalies, was used for the anomaly detection task. To address the comments from the reviewers, especially reviewers 99iS and znEu, we are currently replicating our experiments on the 'Sepsis' dataset from The PhysioNet/Computing in Cardiology Challenge 2019, which contains multivariate data sourced from 40 000 Intensive Care Unit patients in three separate hospital systems. We chose this dataset because it contains real-life data, includes missing data, and anomalie appear by segment. Link of the dataset can be found here: https://journals.lww.com/ccmjournal/fulltext/2020/02000/early_prediction_of_sepsis_from_clinical_data__the.10.aspx.
>  We will have the results shortly, providing us with a second dataset for anomaly detection.
>
> **Q2. How does the issues of similarity/dissimilarity play out in multivariate settings? What if the switching behaviour is only present in a few of the channels?**
>
> **A2.** This issue of similarity and dissimilarity in contrastive learning is treated the same way for univariate and multivariate settings. The semantic choice of which instances should be similar/dissimilar is entirely dependent on the method developed and the problem at hand. For example, in time-series it is common to want to represent two samples from nearby timesteps similarly. However, the way in which the similarity/dissimalirity of representations is evaluated is the same for univariate/multivariate: a dot product is computed between two representation vectors as a measure of similarity.
>
> With regards to the state switching issue, this happens regardless of the number of affected channels. The issue with contrastive learning and finite-state systems is the pair selection process: in the introduction of our paper, we explain that contrastive learning tasks “define positive pairs by proximity in time, and negative pairs by points that are distant in time (Banville et al., 2021; Franceschi et al., 2019), which can incur sampling bias issues. Points of a negative pair might be far in time but close to a period apart (i.e. very similar) and points of a positive pair might be close but very different (think of a pulsatile signal for example).” Whether the switching state behaviour is present in a few or all channels, this information will be lost during training because of the pair selection process.
>
> **Q3. Why does T-Rep perform better in some cases and not it others? For instance, in Table 5 in ETTh_2 T-Rep performs better at larger horizons, while being significantly worse at lower ones. This is counter-intuitive, since in section 4.2.2. the paper mentions that the pretext forecasting task specifically aims to predict shorter horizons.**
>
> **A3.** We appreciate this question, which we also considered during our experiments. We lack both theoretical and experimental evidence for why T-Rep performs better on longer horizons. However, our intuition leads us to believe that on a short horizon, we are still within a Markovian process, with the major dependence on the preceding moment. On a longer horizon, periodic processes come into play, and the model excels through its handling of time embeddings. This, however, remains speculative, and extensive experiments will need to be conducted in future work to confirm this.
>
> **Q4. It seems that the methods requires supervision in pretest task 4.2.1 and not in 4.2.2 for the pretext task, can the authors comment on potential use of T-Rep in completely unsupervised pretraining?**
>
> **A4.** It may not be expressed clearly enough in the article, but we confirm that both pretext task (4.2.1 and 4.2.2) are supervised. T-Rep as a whole is trained on datasets without labels (there is an ‘X’, but no ‘y’), so the training of this model can be regarded as unsupervised. However, it relies on the creation of artificial supervisory signals constructed from the given ‘X’. These artificial supervisory signals are what we call ‘pretext tasks’, and do not require any data or information beyond the given dataset ‘X’. This duality between not using any labels for the dataset but creating artificial supervised signals (=pretext tasks) to train the model resulted in the field’s name, “Self-supervised learning”: the supervision in pretext tasks originates from the input data.

---

> > ### Author Response · Authors · 2023-11-16
> > **Response to Reviewer rWLt - Part 2**
> >
> > **Q5. How much overlap do we need for contextual consistency? How is this parameter decided?**
> >
> > **A5.** The overlapping window is not a parameter of the model, the overlapping subseries are randomly sampled. We kindly invite you to consult the article TS2Vec: Towards Universal Representation of Time Series by Yue et al. (https://ojs.aaai.org/index.php/AAAI/article/view/20881) who are the inventors of contextual consistency.
> > We regularly cite this article in our submission, particularly at the beginning of paragraph 3 when we say *These innovations are combined with state-of-the-art methods for spatial feature-learning and model training, the contextual consistency and hierarchical loss frameworks (Yue et al., 2022).*
> > They explain it well in the subsection called **Random Cropping**. We can include their explanations in our appendix to make things easier for readres if you think it's necessary.
> >
> > Please note that we can't modify this parameter in our ablation study, as it isn't a parameter of the model.
> >
> > **Q6. In section 3, the terms such as contextual consistency and hierarchical loss are only defined intuitively. These need to be defined in terms of their mathematical formulation(s). Currently, these are difficult to understand. Furthermore, it is also unclear how this plays out mathematically in linear projection layer.**
> >
> > **A6.** Thank you for bringing up this issue, which prompted a lot of reflection during the article writing process. We considered that as contextual consistency and hierarchical loss are not our own innovations but part of the literature through the article 'TS2Vec: Towards Universal Representation of Time Series' by Yue et al. it was not necessary to provide the mathematical formulations. That being said, thanks to your comment we have decided to include these mathematical details in our appendix to save readers some time.
> >
> > **Q7. What is the mathematical action of TCN layer, and what is its role?**
> >
> > **A7.** We kindly invite you to refer to the article by Bai et al., who were the first to propose the generic temporal convolutional network (TCN) architecture in their paper "An Empirical Evaluation of Generic Convolutional and Recurrent Networks for Sequence Modeling" (https://arxiv.org/pdf/1803.01271.pdf). In our case, its utility lies in feature extraction, serving as an encoder for time series. If you believe it is necessary to elaborate on this point in detail in the article, we can include it in the appendix.
> >
> > **Regarding your concern about results not repeated across random seeds:** instead of fixing a seed, which has the disadvantage of not considering the variance that may exist between two seeds, we decided to run each experiment between 10 and 20 times to account for variance and obtain reliable results. The reported results are the average of these runs. This approach maximizes the reliability and robustness of our results, rather than cherry-picking the seed that yields the best results. During our 10 or 20 runs, the seed is set to "None." When the seed is set to “None”, PyTorch utilizes the system time or another unpredictable source to initialize the random number generator. Consequently, each time you run the program, you are likely to get different random numbers.
> >
> > *If our responses and revisions meet your expectations, we would greatly appreciate it if you could consider revising your evaluation accordingly. Thank you for your time and thorough review.*

---

> > > ### Author Response · Authors · 2023-11-22
> > > **Response to Reviewer rWLt - Experiments**
> > >
> > > We have now conducted the ablation study you requested, which you can find in section 5.5 of the paper in blue.
> > > The results confirm that the proposed pretext tasks and the addition of a time-embedding module to the encoder contribute to T-Rep’s performance: removing any of these decreases the scores in both tasks. These results also illustrate the interdependency of both tasks, as in forecasting, only leaving one of the tasks obtains worse results than removing both pretext tasks. It also justifies our preferred choice of time-embedding, since Time2Vec (Kazemi et al., 2019) outperforms the other 2 architectures in both tasks.
> > >
> > > We have also conducted the visualizations you requested, available in section A.1 of the paper in blue. This experiment truly highlights the versatile nature of T-Rep, improving the quality of representations at both the dataset scale (inter-instance discrimination) and individual time series (timestep granularity).
> > >
> > > We modified Fig. 1, as the yellow against a white background was difficult to read, as you mentioned. We replaced yellow with dark orange.
> > >
> > > Finally, we added the mathematical formulations of the contextual consistency and hierarchical loss, as you requested. You can find these formulations in sections A.5 and A.6.
> > >
> > > Thank you for proposing these improvements.
> > >
> > > *If our adjustments meet your expectations, we would greatly appreciate it if you could consider revising your evaluation accordingly. Thank you for your time and thorough review.*

---

### Official Review · Reviewer_99iS · 2023-11-09

**Soundness:** 3 good
**Presentation:** 3 good
**Contribution:** 3 good
**Rating:** 5
**Confidence:** 4

**Summary:**

The paper presents a self-supervised way of learning latent representations of variable length time series data that are useful for various downstream tasks like time series anomaly detection, forecasting and classification. The authors propose two surrogate loss functions in form of 'pretext tasks' that influence the learned representation/embedding to persist temporal consistency in latent space along with information to forecast, and thus be robust to missing data. The results are demonstrated on three downstream tasks where T-Rep outperforms the SOTA time series representation learning methods.

**Strengths:**

Below are the strengths of this proposed work:
1. The problem is highly relevant to the time series research community and well motivated by the authors. They also do a good job at covering the related work and highlighting the necessity of temporal robustness in the representations that's usually overlooked when learning time series representation.
2. The authors introduce surrogate loss functions to train the representation model in a self supervised manner which they refer to as 'pretext tasks'. The two loss functions (although not novel) are practically reasonable given the need of temporal consistency and predictability in the learned representations.
3. Authors do a good job of testing T-Rep across three key downstream tasks which helps in validating the usefulness of learned representation as task-agnostic.

**Weaknesses:**

Following are weaknesses of this work:
1. The effectiveness of the proposed method relies on the evaluation done mainly on UCR/UEA and Yahoo Datasets which are infamous for their incorrectness, and triviality in labels. Refer: https://arxiv.org/pdf/2009.13807.pdf. This questions the effectiveness of the proposed method and how well would it work in real-world scenarios?
2. Similar to Pt.1, I feel the downstream benchmarks are bit too trivial to test the true effectiveness/usefulness of a representation learning method. For e.g., in the task of anomaly detection, the authors propose to use a simple point-based evaluation method which is not realistic, as real world anomalies are usually segment based and need more rigorous evalution to demonstrate usefulness. Refer: https://arxiv.org/pdf/2109.05257.pdf.
3. In Sec 5.5, I get that one can manually look at the heatmaps to infer similarities between the patterns in raw time series signal and the learned representation but I didn't fully understand the usefulness of that? Can I just look at the heatmap and use that information? if so, how? Manually matching pattern is pleasing but I don't see any usability, and hence don't see any true interpretability coming out of it.

**Questions:**

My questions can be found in Weaknesses section.

---

> ### Author Response · Authors · 2023-11-16
> **Response to Reviewer 99iS**
>
> We appreciate your insightful review of our paper. We have diligently addressed each of your comments point by point, aiming to enhance the overall quality of our work in response to your valuable feedback.
>
> **Q1. The effectiveness of the proposed method relies on the evaluation done mainly on UCR/UEA and Yahoo Datasets which are infamous for their incorrectness, and triviality in labels. Refer: https://arxiv.org/pdf/2009.13807.pdf. This questions the effectiveness of the proposed method and how well would it work in real-world scenarios?
> Q2. Similar to Pt.1, I feel the downstream benchmarks are bit too trivial to test the true effectiveness/usefulness of a representation learning method. For e.g., in the task of anomaly detection, the authors propose to use a simple point-based evaluation method which is not realistic, as real world anomalies are usually segment based and need more rigorous evalution to demonstrate usefulness. Refer: https://arxiv.org/pdf/2109.05257.pdf.**
>
> **A1 and A2.** We acknowledge the concerns you raised regarding the datasets used for anomaly detection evaluation and the behavior of the model in real-world scenarios. We are aware of the limitations of these datasets, and we have recently become acquainted with them.  We completely agree with the necessity of working with less trivial datasets.
>
> To address those 2 questions, we are currently testing our model on the 'Sepsis' dataset from The PhysioNet/Computing in Cardiology Challenge 2019, which contains multivariate data sourced from ICU patients in three separate hospital systems. We chose this dataset because it contains real-life data, includes missing data, and anomalies appear by segment. We are, in fact, implementing the PA%K protocol presented in the article you suggested (Towards a Rigorous Evaluation of Time-series Anomaly Detection, Kim et al., https://arxiv.org/pdf/2109.05257.pdf.), which helps alleviate the overestimation effect of point-based evaluation methods.
> This allows us to address both your concerns regarding the Yahoo Dataset while demonstrating that our T-Rep model is effective in real-world scenarios with messy data.
> Link of the dataset can be found here: https://journals.lww.com/ccmjournal/fulltext/2020/02000/early_prediction_of_sepsis_from_clinical_data__the.10.aspx
> We will communicate our results to you as soon as possible and include them in the article
>
> **Q3. In Sec 5.5, I get that one can manually look at the heatmaps to infer similarities between the patterns in raw time series signal and the learned representation but I didn't fully understand the usefulness of that? Can I just look at the heatmap and use that information? if so, how? Manually matching pattern is pleasing but I don't see any usability, and hence don't see any true interpretability coming out of it.**
>
> **A3.** What this heatmap shows is two-fold.
>
> Firstly, it serves as a valuable sanity check by offering a qualitative means of verifying the preservation of information and properties from the original signal in the representations. To give an example, this can be used by a data scientist working on a change point detection model to see if there's an abrupt change in the probability distribution of the signal.
>
> Secondly, it shows that the features extracted by the model are interpretable: we can visually see that variations in the representation vectors correspond to human-interpretable features such as periodicity, variance, anomalies, distribution shifts etc. One could imagine that the model learns features which aren’t human interpretable. For instance, in a CNN, the first few layers tend to capture high-level features like edges or textures, but the last few layers learn more abstract and complex features, which we can’t visually make sense of. It is very positive to us to see that this is not the case with T-Rep and that the extracted features can be visually interpreted and matched with properties of the original signal. Whilst we agree one shouldn’t rely on visual pattern matching, this provides advantages by providing visual support. We agree this visual aid is not sufficient for downstream tasks, where we rely on comparing representations quantitatively rather than qualitatively.
>
>
>
> *If our responses and revisions meet your expectations, we would greatly appreciate it if you could consider revising your evaluation accordingly. Thank you for your time and thorough review.*

---

> ### Author Response · Authors · 2023-11-22
> **Response to Reviewer 99iS - Experiments**
>
> We have now conducted the experiment you requested, which you can find in section 5.1 of the paper in blue.
> The results on the Sepsis dataset confirm that T-Rep performs better than TS2Vec and a linear baseline on a dataset different from UCR/UEA and Yahoo Datasets. The proposed method has a great effectiveness in a real-world scenario and on non-trivial downstream benchmarks, even when using an evaluation method that is not point-based.
> We have also added the two references you suggested.
>
> Thank you for proposing this improvement.
>
> *If our adjustments meet your expectations, we would greatly appreciate it if you could consider revising your evaluation accordingly. Thank you for your time and thorough review.*

---

### Author Response · Authors · 2023-11-16
**Common response to all reviewers**

Thank you sincerely for the meticulous review of our submission. We highly appreciate the time and effort you have invested in providing such insightful and constructive comments. Your thorough examination has undeniably played a pivotal role in elevating the quality of our paper.

We are currently working on the implementation of several experiments requested by various reviewers. Some of them may be completed by the end of the discussion period, while others will take more time. However, we commit to including all of them in the final version of the article because the setup for these experiments has already begun—it's just a matter of computation time. We are implementing the following additional experiments:

1. **Ablation Study:** We completely agree that this element is necessary to attest to the quality of our model. This ablation study includes removing pretext tasks one by one, as well as the time-embedding module. We evaluate these model variations in forecasting and anomaly detection tasks - they are most likely to be affected by such changes, as they required time step-granularity representations, the core focus of T-Rep.
Having started the ablation study, we can already provide you with results of the ablation study on the forecasting task. We observe that the two pretext tasks complement each other well: if only one of them is used, the performance is lower than using none, and significantly lower than using both. Additionally, we can see that the time-embedding (TE) module effectively enhances performance, and it is the combination of these elements (TE + 2 proposed pretext tasks) that allows us to outperform the state-of-the-art TS2Vec.

|                                   | Forecasting |
|-----------------------------------|-------------|
|                                   | Avg. MSE    |
|-----------------------------------|-------------|
| **T-Rep**                          | **0.986**   |
|-----------------------------------|-------------|
| *Pretext tasks*                    |             |
| w/o TE-conditioned forecasting    | 1.022 (+3.7%) |
| w/o TE divergence prediction       | 1.003 (+1.7%) |
| w/o New pretext tasks               | 0.999 (+1.3%) |
|-----------------------------------|-------------|
| *Architecture*                     |             |
| w/o TE module (=TS2Vec)            | 1.004 (+1.8%) |

*Ablation results on ETT forecasting datasets. The percentage changes are calculated as the relative difference between the modified model's performance and T-Rep's performance.*

2. **Latent Space Visualization and Unsupervised Evaluations:** Several of you rightly pointed out that adding unsupervised evaluations like clustering could help assess representation quality and so, we will add experiments in this direction.

3. **New and Complex Anomaly Detection Dataset:** The 'Sepsis' dataset from The PhysioNet/Computing in Cardiology Challenge 2019, which contains multivariate data sourced from ICU patients in three separate hospital systems. We chose this dataset because it contains real-life data, includes missing data, and anomalies appear by segment. This allows us to address both concerns regarding the Yahoo Dataset while demonstrating that our T-Rep model is effective in real-world scenarios with messy data and anomalies appearing by segment, thus not relying solely on a point-based evaluation.

4. **New Baselines for Forecasting and Classification:** As you suggested in order to enhance the quality of the article.

These experiments are underway. While waiting to present them to you, we addressed all your questions, aiming to clarify points that lacked clarity in our initial submission (please see each individual response).

---

> ### Author Response · Authors · 2023-11-22
> **Rebuttal submitted**
>
> Dear reviewers,
> We have finished taking into account all your comments.
> You can find the updated version of our article, with all the modifications we made highlighted in blue.
>
> These modifications include:
>
> - Improving the colors of Figure 1 to enhance visibility
> - Adding experiments on the Sepsis dataset for segment-based anomaly detection on a real-life dataset
> - Introducing the Minirocket baseline for the classification task
> - Adding a linear baseline for the forecasting task
> - Conducting a comprehensive ablation study
> - UMAP visualization of learned representations from T-REP and TS2VEC
> - Mathematical formalizations of contextual consistency and hierarchical loss
>
> We believe that these additions significantly enhance the quality of our submission, and we sincerely thank you for guiding us through their implementation.
>
> In addition to these various additions, we have addressed all of your questions. Please see each individual response for more details.

---

### Meta-Review · Area_Chair_HdJe · 2023-12-07

**Metareview:**

This paper introduces an innovative self-supervised method for time-series representation learning using time-embeddings. While reviewers appreciated the originality of the method and the quality of the experiments, they highlighted the need for improved experimental design and baseline selection. In response, the authors conducted additional experiments and analyses, addressing these concerns effectively. They demonstrated the method's robustness on various datasets and its capability to handle missing data, further strengthening their argument. The authors' comprehensive responses and enhancements to the paper substantiate the recommendation for acceptance, acknowledging its contribution to time series analysis and representation learning.

**Justification For Why Not Higher Score:**

The paper's score reflects its current state, acknowledging its significant contributions while recognizing areas for improvement. The method's originality and effective handling of various datasets are commendable. However, the initial concerns regarding experimental design and baseline comparisons, despite being addressed, indicate room for further enhancement, particularly in comparative analysis with state-of-the-art methods. This balance between the paper's strengths and areas for improvement justifies not assigning a higher score.

**Justification For Why Not Lower Score:**

The paper merits its current score due to its innovative approach in time series representation learning and the authors' thorough response to reviewer feedback. The additional experiments and analyses provided by the authors effectively address the initial concerns, demonstrating the method's robustness and applicability. The paper makes a substantial contribution to the field, warranting its acceptance and precluding a lower score.

---

### Decision · Program_Chairs · 2024-01-16

Accept (poster)